# Uncritical tokens are 'critical' in pretraining: the implicit regularization effect of Next token prediction

## Abstract

Next Token Prediction (NTP) is the prevailing pre-training approach for large language models, which have demonstrated remarkable reasoning capabilities. A key characteristic of NTP is its objective to predict every token in a sequence, including tokens that are not directly relevant to the final answer or core logic—often considered training noise. While such "noise" from uncritical tokens is traditionally thought to impair learning by introducing irrelevant information, our research reveals a counterintuitive positive effect. To isolate this phenomenon, we contrast NTP with Critical Token Prediction (CTP), a training paradigm that focuses exclusively on specific tokens such as the final answer. Our findings show that NTP consistently surpasses CTP in reasoning ability. We hypothesize and substantiate through theoretical analysis that the learning objective on uncritical tokens acts as an implicit regularizer, analogous to explicit $L^2$ regularization. Further empirical analysis across various benchmark reasoning datasets confirms that NTP-trained models exhibit enhanced generalization and robustness, demonstrating greater resilience to perturbations and achieving flatter loss minima. These findings reveal that uncritical tokens are, in fact, 'critical' for developing robust reasoning during pre-training, offering valuable insights into optimizing training strategies for LLM development.

## 1 Introduction

As transformer-based Large Language Models (LLMs) continue to fuel enthusiasm for Artificial General Intelligence (AGI), numerous techniques are emerging to advance this trend, fostering a highly optimistic outlook for the eventual realization of AGI. A central challenge since the inception of LLMs has been how to efficiently train these models to achieve superior reasoning capabilities. Over time, a series of training techniques have revolutionized the performance of LLMs, each contributing to significant milestones in the field.

The success of natural language processing (NLP) has been significantly driven by the widespread adoption of next token prediction (NTP), a self-supervised learning approach popularized by the GPT series (Radford & Narasimhan, 2018; Radford et al., 2019; Brown et al., 2020). Unlike supervised methods that depend on costly labeled data, NTP enables models to learn from vast amounts of unlabeled text by predicting subsequent tokens, allowing for zero-shot generalization and eliminating the need for task-specific fine-tuning. This framework has established NTP as a cornerstone of modern NLP.

In contrast to NTP, supervised training only on labels can be regarded as critical token prediction (CTP), illustrated in Fig. 1. Although NTP has been successfully applied in LLMs, it still leaves room for speculation: Given the availability of labeled data, should CTP be reconsidered as a viable alternative? For instance, in training a model for arithmetic addition, employing NTP to learn problem formulations seems inherently flawed, as the subsequent components cannot and should not be inferred from preceding ones in math problems. Furthermore, recent advancements have increasingly focused on the strategic selection of important tokens for training. For example, RHO-1 (Lin et al., 2025a) utilizes a model to score each token and trains only on samples with high scores. Phi-4 (Abdin et al., 2024) has made significant strides in enhancing reasoning capabilities through

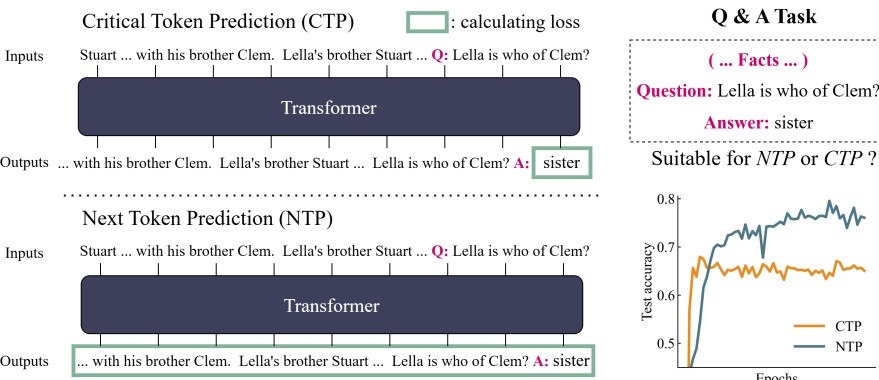

Figure 1: Schematic illustration and test performance comparison of NTP and CTP on the CLUTRR task. Regarding their training objectives, especially for tasks like arithmetic addition, CTP's loss function exclusively focuses on the answer, while NTP's loss encompasses the entire sequence. This difference introduces implicit noise during NTP's optimization process.

an emphasis on data quality. One key technique involves synthesizing a large number of question and answer (Q&A) data pairs, even during the pretraining phase with NTP. This raises a natural question: since the answer portion of Q&A data can be seen as a form of label, should CTP be used for Q&A pairs instead?

In this study, we conduct a systematic comparison between NTP and CTP using composition tasks and generalizing to more realistic reasoning tasks. **Our empirical and theoretical findings reveal that NTP, in specific circumstance, is a variance of weight decay regularization, due to the noise inherent in training set.** To further investigate this bias, we employ series of realistic tasks, especially for multi-hop reasoning task PrOntoQA, providing additional empirical evidence. Beyond the regularization, we observe that models trained with NTP demonstrate greater robustness and flatness than those trained with CTP.

## 2 RELATED WORK

**Next-Token Prediction and Other Training Methods.** Next token prediction (NTP) is a widely used method for training LLMs. Recent studies analyze NTP from various angles, investigating geometric properties in logits space (Zhao et al., 2024; Thrampoulidis, 2024), theoretical capacity in transformers (Madden et al., 2024), mechanistic insights (Li et al., 2024), and empirical scaling laws (He & Su, 2024). Recognizing NTP's limitations, alternative training paradigms have emerged (Bachmann & Nagarajan, 2024; Gloeckle et al., 2024; Lin et al., 2025b; Havrilla & Iyer, 2024). For example, RHO-1 (Lin et al., 2025a) introduces a token-level scoring mechanism, selectively training on high-scoring samples to improve efficiency. Similarly, Phi-4 (Abdin et al., 2024) demonstrates significant advancements in reasoning capabilities by prioritizing high-quality data during training, and (Huerta-Enochian & Ko, 2024) presents the study analyzing the effects of various prompt loss token weights for supervised finetuning. Despite these advances, the link between diverse training methods and generalization remains underexplored. A deeper understanding of this relationship is crucial for advancing the field and developing more robust and efficient LLMs.

**Implicit Bias for Noise-Induced Regularization Techniques.** Implicit bias from noise-induced regularization is widely studied, with different noise forms impacting training and performance (Zhu et al., 2019). Stochastic gradient descent (SGD) noise is a key example, shown to improve generalization by promoting flatter loss landscapes (Wu et al., 2020; Feng & Tu, 2021; Xie et al., 2020), with its magnitude depending on the landscape (Mori et al., 2021) and linked to dynamical stability (Wu et al., 2018; Ma & Ying, 2021). Dropout is another common technique enhancing generalization (Zhang et al., 2022; Zehui et al., 2019; Zhou et al., 2020; Li et al., 2023; Fan et al., 2019; Wu et al., 2021; He et al., 2024), with studies showing its noise improves generalization from various perspectives (Mianjy et al., 2018; Bank & Giryes, 2020; Lengerich et al., 2022; Cavazza

et al., 2018; Wei et al., 2020; Zhang et al., 2023b), including fostering condensation and improving loss landscape flatness (Zhang & Xu, 2024). In this work, we draw an analogy between NTP and noise-induced training methods to explore NTP's impact on model reasoning capabilities.

## 3 PRELIMINARIES

In this section, we introduce some key definitions of the training methods (NTP & CTP) and the synthetic task setup. The detailed definition could be referred to Appendix D.

### 3.1 DEFINITION OF NTP AND CTP

We note the input sequence with length $T$ in the token format $\{x_k\}_{k=1}^T$, and without loss of generality, the critical token is set as the end token $x_T$. We also denote $P_{\boldsymbol{\theta}}(x_{t+1}|x_{\leq t}) = P_{\boldsymbol{\theta}}(x_{t+1}|x_1, \ldots, x_t)$ as the model output logits at the $t$-token. Training loss of NTP and CTP are defined as follows:

$$\mathcal{L}_N = -\frac{1}{T}\sum_{t=1}^{T-1} 1\{x_{t+1}\}\log(P_{\boldsymbol{\theta}}(x_{t+1}|x_{\leq t})), \tag{1}$$

$$\mathcal{L}_C = -1\{x_T\}\log(P_{\boldsymbol{\theta}}(x_T|x_{\leq T-1})). \tag{2}$$

Not hard to see that the CTP loss $\mathcal{L}_C$ is the critical part of the NTP loss $\mathcal{L}_N$, only calculated on the critical token $x_T$. We use the original GPT-2 structure and denote $\mathrm{Attn}$ and $\mathrm{MLP}$ as the attention and fully connected block separately.

### 3.2 THE ANCHOR FUNCTION TASK SETUP

The anchor function (Zhang et al., 2024b) is designed to cook synthetic dataset that can mimic language tasks but provides a clear examination of the model's performance and mechanisms on compositional functions.

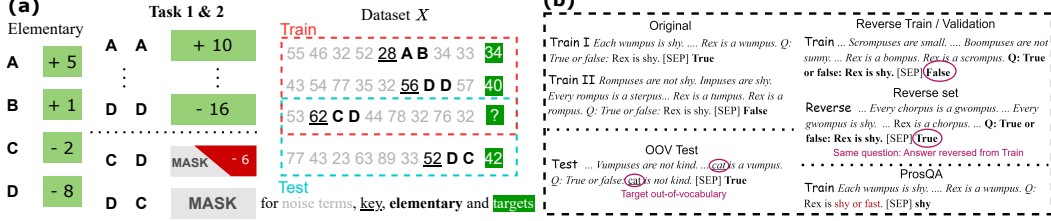

Figure 2: Illustration of the two primary tasks discussed in this work. (a) the Anchor Function task and (b) the PrOntoQA tasks. **(a):** The Anchor Function task setup. The composition $(C, D)$ is used as a test example in Task 1 and as a training example with a misleading $x - 6$ operation in Task 2. We focus on the model preference on the symmetric pair $(D, C)$, which is excluded from training. A prediction matching the elementary anchor composition rule (i.e., $x - 10$) is considered a reasoning solution; otherwise non-reasoning solution. **(b):** The description of different PrOntoQA tasks used in this work: Original, reverse, OOV test and its variation ProsQA (Hao et al., 2024).

**Definition of anchor function** Consider a function $g(\mathbf{x}) : \mathbb{R}^{s \times d} \to \mathbb{R}^C$, where $s$ represents for sequence length while $C$ for vocabulary size. The input $X$ consists of two parts: anchor set $\mathcal{A} = \{A, B, C, D\}$ and the definition domain of function $f$, $\mathcal{D} = \{20, \ldots, 100\}$. The function is defined as:

$$g(\ldots, x_i, a, x_{i+2}, \ldots) := a(x_i), \quad \text{while } a \in \mathcal{A}; x_i \in \mathcal{D} \tag{3}$$

$$g(\ldots, x_i, a, b, x_{i+3}, \ldots) = (a, b)(x_i) := b(a(x_i)), \quad \text{while } a, b \in \mathcal{A}; x_i \in \mathcal{D}. \tag{4}$$

The latter setting is also called a composition task. In this work, we set the specific elementary functions $f_a, f_b$ as:

$$A(x) = x + 5, \quad B(x) = x + 1, \quad C(x) = x - 2, \quad D(x) = x - 8.$$

The anchor function operates solely on the position preceding the anchor $a$, which is denoted as the key item, and is independent of the input at other positions.

**Two versions of the composition task**   To rigorously assess the reasoning capabilities of models, we have designed two composition tasks with escalating levels of difficulty.

Task 1: We remove the pairs $(C, D)$ and $(D, C)$ from a set of 16 possible combinations of anchor pairs, thereby withholding direct information about these specific compositions from the model.

Task 2: We remove the pair $(D, C)$ and introduce misleading information by presenting $(C, D)(x)$ as $x - 6$, despite the correct operation being $(C, D)(x) = x - 10$.

Here we define that the reasoning solution as the model that can learn the function of $D$ and $C$ respectively. If the model fails to identify elementary functions, we call it a non-reasoning solution. For both tasks, we evaluate the model's reasoning ability by measuring its accuracy in determining that $(D, C)(x) = x - 10$. This assessment is critical because it requires the model to discern the roles of the elementary anchors $\mathcal{A}$ and correctly compose their operations. Only by accurately identifying the functions of these elementary anchors can the model successfully address the composition problem, analogous to human reasoning processes. See Appendix D.1 for details on model architecture and data generation.

## 4   REGULARIZATION: IMPLICIT REGULARIZATION FROM NTP

An interesting question is, whether the transformer could learn the elementary functions only with composite functions. In this section, we analyze the NTP-trained and CTP-trained models on composition tasks. We also provide a comparison with CTP + large weight decay experiments.

### 4.1   EFFECTS OF NTP TRAINING

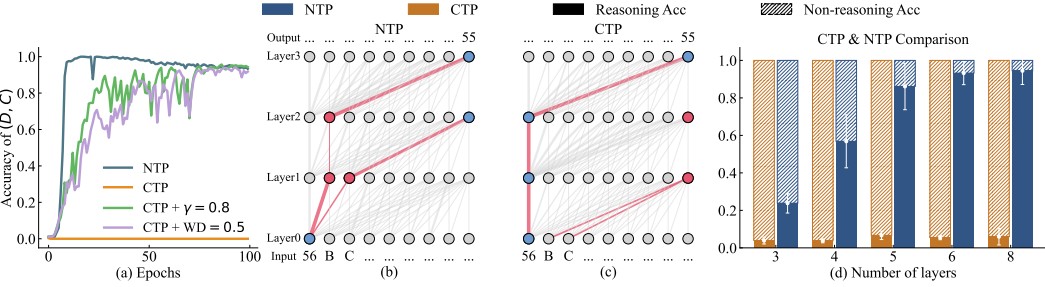

Figure 3: Accuracy of masked pair $(D, C)$ of composition task. The *NTP* and *CTP* represent the normal setting ($\gamma = 0.5, \mathrm{WD} = 0$) . **(a):** 3-layer 1-head GPT-2 model performance on **task 1**, NTP achieves similar reasoning ability compared with regularization methods like small initialization or weight decay, which theoretically discussed in Theorem 1. **(b & c):** The information flow of the composition pair $(B, C)$ of the NTP-trained model and CTP-trained model in (a). The NTP-trained model treats the anchors one by one, while the CTP-trained model merges the anchors in Layer 0 and finishes it in Layer 3. This is a shortcut learning pattern and indicates the CTP's failure in Task 2. **(d):** The non-reasoning and reasoning solution of **task 2** with different layers. The NTP could stably switch the non-reasoning solution to the reasoning solution. The error bars represent the standard deviation across 3-time runs on GPT-2.

In the (Zhang et al., 2024a; 2025), authors have figured out that the initialization scale will affect the preference of the model. They initialize the layer in a normal distribution $\mathcal{N}\left(0, (d_1)^{-2\gamma}\right)$ with $d_1$ input neurons and initialization scale $\gamma$. The smaller initialization (large $\gamma$) of parameters contributes to a more generalized model, while the large initialization (small $\gamma$) will lead to poor performance in both ID (in distribution,like $(A, B)$) and OOD tasks (out of distribution, like $(D, C)$). The authors find that the watershed of reasoning and non-reasoning is the $\gamma = 0.5$, which is similar to default Kaiming normal initialization. However, the authors only focus on CTP training.

Here we establish that, with the Kaiming normal scale in which transformer should select the non-reasoning solution, which will be shifted to a reasoning solution using NTP rather than CTP. From Fig. 3, for task 1, NTP-trained models could learn pairs $(C, D)$ and $(D, C)$ from the rest, however, CTP-trained models cannot figure out, but mistakenly induce $(D, C)$ as the same of $(D, D)$. For task 2, after introducing the misleading $(C, D)$, CTP remains focused on the unreasoning solution, while NTP prefers reasoning solutions, and this tendency becomes increasingly evident as the depth of the model increases.

We leverage the information flow analysis to reveal the mechanism behind the reasoning or non-reasoning solutions. The information flow is about the NTP-trained and CTP-trained for a three-layer model of GPT-2 with the same testing sample. The thickness of the line connecting the $j$-th token in Layer $l$ and the $k$-th token in Layer $l + 1$ represents attention score at position $(k, j)$. For reasoning solution, the model treats two anchors one-by-one, which aligns with the original intent of the composition task design. In contrast, CTP selects a shortcut to fit the train set: Considering the symmetric property of the data $((a, b)(x) = (b, a)(x)$ for all $a, b \in \mathcal{A})$, it merges two anchors in the first layer and then treats the combination of anchors as the new. This approach represents a trade-off wherein the model learns the combination rather than elementary anchors, enabling it to effectively fit the training set with only a two-layer network. However, the drawback is evident: when faced with $(C, D)$ combination that was absent from training, the model fails.

eight decay is another widely used regularization technique that aids generalization, and boosts the reasoning ability shown in Fig. 3(a). In our settings, incorporating weight decay with CTP training could prevent the model from exhibiting shortcut learning, a phenomenon observed in pure CTP training. In next section, we will establish the theoretical analysis on the effect of NTP training.

## 4.2 THEORY ANALYSIS OF NTP TRAINING ON COMPOSITION TASK

To explore the different training results of NTP and CTP, we need to carefully analyze the loss $\mathcal{L}_N$ and $\mathcal{L}_C$. In this section, we start with general theory and then applies it to two training phases to illustrate the regularization effect of NTP. First, we state the regularization effect in the initial stage as follows.

**Theorem 1** (NTP regularization). *Suppose there exists a parameter vector $\boldsymbol{\theta}_0$ such that, for every uncritical position $X = (x_1, \ldots, x_t)$ with $t < T - 1$, such that $P_{\boldsymbol{\theta}_0}(\cdot \mid X) = \frac{1}{d_{\mathrm{vob}}} \mathbf{1}_{d_{\mathrm{vob}}}$. Then, for $\boldsymbol{\theta}$ in a neighborhood of $\boldsymbol{\theta}_0$, the NTP loss admits following expansion as sample size tends to infinity,*

$$\mathcal{L}_N(\boldsymbol{\theta}) = \frac{1}{T} \mathcal{L}_C(\boldsymbol{\theta}) + \frac{1}{2} (\boldsymbol{\theta} - \boldsymbol{\theta}_0)^\intercal I_{\boldsymbol{\theta}_0} (\boldsymbol{\theta} - \boldsymbol{\theta}_0) + \mathcal{O}\big(\|\boldsymbol{\theta} - \boldsymbol{\theta}_0\|^3\big), \tag{5}$$

*where $I_{\boldsymbol{\theta}_0}$ denotes the empirical Fisher information matrix at $\boldsymbol{\theta}_0$ which is*

$$I_{\boldsymbol{\theta}_0} = \frac{1}{T} \sum_{t=1}^{T-2} \mathbb{E}_{X_t} I_{\boldsymbol{\theta}_0}(X) \tag{6}$$

Next, we apply this theorem to the two training phases. To do so, we first introduce an abstraction of the sequence model. Typically, in such models, an input sequence $X$ undergoes various transformations before being mapped to logits via a projection matrix. We formalize this process as follows:

**Definition 1.** *Let $X$ be the embedded input sequence, the model generates the logit vector*

$$\ell(X) = G_{\hat{\boldsymbol{\theta}}}(X) W_{\mathrm{proj}} \tag{7}$$

*and the corresponding probabilities*

$$P_{\boldsymbol{\theta}}(X) = \mathrm{softmax}(\ell(X)), \tag{8}$$

*where $\boldsymbol{\theta} = (\hat{\boldsymbol{\theta}}, \mathrm{vec}(W_{\mathrm{proj}}))$.*

Next, we give the explicit expression for regularization in the initial stage.

**Proposition 1** (Regularization in initial stage). *Consider sequence models defined as Def. 1, there exists $\lambda > 0$ such that Eq. (5) has following reformulation at initialization:*

$$\mathcal{L}_N = \frac{1}{T} \mathcal{L}_C + \frac{1}{2} \frac{\lambda}{d_{vob}} \mathrm{vec}(W_{\mathrm{proj}})^\intercal \left( I - \frac{1}{d_{vob}} \mathbf{1} \mathbf{1}^T \right) \otimes I \, \mathrm{vec}(W_{\mathrm{proj}}) + \mathcal{O}(\|\boldsymbol{\theta}\|^3)$$

This Proposition reveals the close relationship between NTP and $L^2$ regularization. It proves that, in the early stages of training, NTP can be considered a special type of $L^2$ normalization, which explains why models trained with NTP perform similarly to models trained with CTP and weight decay, as shown in Fig. 3 (a). The proofs could be find at Appendix B.1.

Finally, we conclude this section with a discussion of the final convergence phase. At the end of the training, since the rest of the sequence (excluded critical tokens) is uniform noise, the logits for non-critical tokens should converge to a uniform distribution. Based on the above insight, Theorem 1 still applies to the analysis of the final state which suggests that NTP training will select a flatter solution.

**Remark 1.** *Distinction from Standard Label Noise and SGD. The fundamental difference from standard analyses of label noise or SGD lies in the structure of the noise. In CTP with label noise, perturbations are restricted solely to the critical token's loss (affecting only $f_S(\theta, x)$). In contrast, NTP introduces noise across the entire sequence, affecting the function outputs at every token position ($f_s, \forall s \in [0, S]$).*

**Remark 2.** *Intuitive Sketch of the Quadratic Term in Thm.1 Our derivation considers the local behavior of a parametric model where both input and output distributions are assumed to be uniform. By strictly applying the Taylor expansion of the KL divergence, we have:*

$$\mathrm{KL}(P_{\theta_0} \| P_\theta) = \frac{1}{2}(\theta - \theta_0)^\intercal I_{\theta_0}(\theta - \theta_0) + \mathcal{O}(\|\theta - \theta_0\|^3),$$

*where $I_{\theta_0}$ denotes the Fisher Information Matrix at $\theta_0$. A key insight here is that the first-order term vanishes automatically due to the fundamental property of the score function (i.e., $\mathbb{E}[\nabla \log p] = 0$), leaving the quadratic form as the dominant term. At initialization, expanding around the origin relates this term closely to standard $L_2$ regularization.*

## 5 REALISTIC EXPERIMENTAL RESULTS

When discussing the reasoning capabilities of LLMs, language inference and multi-hop question answering tasks should be taken into consideration (Yu et al., 2024). As an example shown in Fig. 1, these tasks are typically structured in a question-answer format, making them particularly well-suited for training using CTP, where the loss is calculated only on the answer tokens. Alternatively, one can employ NTP and compute the loss for the entire sentence. To eliminate interference from pretrained models, all our experiments are conducted by **training the models from scratch**.

### 5.1 PRONTOQA TASK

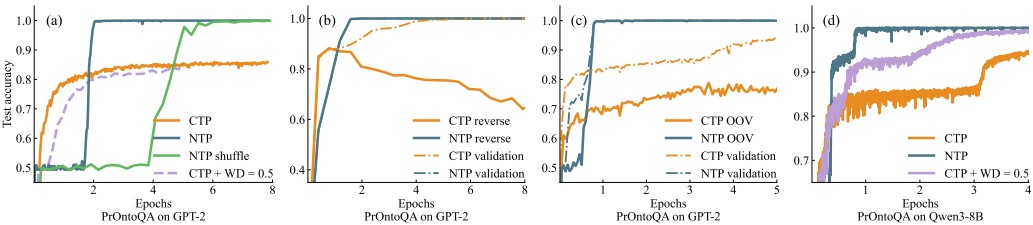

Figure 4: NTP and CTP performance on PrOntoQA tasks. **(a):** Accuracy of NTP and CTP on the original 2-hop PrOntoQA task over training epochs. NTP eventually achieves perfect accuracy, while CTP plateaus around 80%. The NTP shuffle experiment destroys the language structure in facts and query, which are mainly discussed in Appendix A.2. **(b):** 2-hop reverse PrOntoQA: NTP maintains high reasoning accuracy both on validation and reverse test set, but CTP memorizes the training data, leading to decreased accuracy on the reverse test set. **(c):** 1-hop OOV PrOntoQA: NTP achieves nearly 100% accuracy, while CTP stabilizes around 70%. **(d):** The performance of Qwen3-8B model on original 2-hop task.

We conducted comprehensive evaluations on PrOntoQA, a synthetic multi-hop inference dataset with a 50% random guess accuracy. As illustrated with examples in Fig. 2(b), we also proposed two

modified variants of this task specifically designed to better support research on model generalization capabilities.

**Original PrOntoQA task**  Following the experiment established in the (Saparov & He, 2023), we implemented both NTP and CTP on the original PrOntoQA dataset. Both training methods (NTP and CTP) easily surpass the random guess accuracy of 50%. CTP initially learns the mappings effectively but stagnates at around 80% accuracy. In contrast, NTP learns more slowly due to the presence of numerous noise terms but ultimately achieves 100%

Table 1: Comparison of NTP and CTP Performance on modern LLM.

| Model | NTP | CTP |
|-------|-----|-----|
| Qwen2.5-0.5B | **1.00** | 0.87 |
| TinyLlama-1.1B | **0.99** | 0.84 |
| OLMo-1B | **1.00** | 0.91 |

accuracy, exhibiting an accuracy grokking phenomenon as shown in Fig. 4 (a). To assess the robustness of these findings, we further validated this phenomenon on more realistic model architectures. As presented in Table 1, consistent trends were observed across modern large language models (including Qwen2.5, TinyLlama, and OLMo), confirming that the superiority of NTP generalizes beyond the model scale.

**Reverse PrOntoQA task**  In the original PrOntoQA dataset, answers are solely determined by facts within each example; therefore, situations may arise where the same question in different examples is paired with contradictory facts. To better assess the robustness differences between NTP and CTP, we created **reverse** train, validation and test set: For the training and validation sets, we ensured that identical questions have the same answers. For the test set, we modified the facts so that the correct answers are the opposite of those in the training set, which is shown in Fig. 2(b). Therefore during training process, the model has two possible learning paths: 1) learning the reasoning from facts or 2) memorizing all question answers in train set.

On the test set, we observe that while CTP enables the network to achieve an accuracy close to 0.8 initially, it rapidly turns to path 2) and begins to memorize and gradually forgetting the underlying reasoning rules. In contrast, NTP maintains an accuracy close to 1.0 over an extended period both on validation and test, demonstrating strong resistance to overfitting, as illustrated in Fig. 4(b).

**OOV PrOntoQA task**  Based on the original construction of PrOntoQA, we introduce an Out-Of-Vocabulary (OOV) dataset, whose targets in queries are not present in the training set, so they are not trained totally. We evaluated the performance differences between models trained using NTP and CTP. Practically, NTP and CTP struggled with 2-hop reasoning on OOV data, we downgraded the dataset to 1-hop reasoning and replicated the experiments. The results, shown in Fig. 4 (c), indicate that CTP maintains an accuracy of approximately 70%, while NTP achieves nearly 100% accuracy on the OOV dataset. This suggests that CTP is influenced by surface patterns in the data, whereas NTP effectively captures the underlying reasoning rules.

**NTP shuffle experiment**  We also design the NTP shuffle experiment (shown in Fig. 4(a)) to address the advantage of NTP **not** due to the model beginning to develop a better language understanding than their CTP trained counterparts like (Zhang & Hashimoto, 2021) has mentioned. When calculating NTP loss, we shuffle the token order in label of the fact + query part, which ensures that the semantic information from the question does not influence the results. When question tokens are shuffled, NTP requires more training time to recognize the irrelevance of noise. The model initially exhibits prolonged "random guess" behavior in accuracy. However, the core conclusion remains unchanged: After sufficient training, the accuracy of NTP shuffle still significantly surpasses CTP. For the code implementation details, please refer to Appendix A.2.

**Effects of weight decay**  We used the inclusion of weight decay as a control baseline to analyze its differences from NTP. We found that while weight decay exhibits a strong regularization effect, this is primarily limited to larger models (e.g., Qwen3-8B). For smaller models such as GPT-2, the inclusion of weight decay has a negligible impact on final performance.

## 5.2 OTHER NATURAL LANGUAGE REASONING TASKS

Except for PrOntoQA, we have meticulously curated a collection of reasoning datasets and implemented necessary preprocessing procedures to ensure data quality and suitability: LogicInference (Ontanon et al., 2022), CLUTRR (Sinha et al., 2019), Ruletaker (Clark et al., 2020), RobustLR (Sanyal et al., 2022), SimpleLogic (Zhang et al., 2023a), PARARULE Plus (Bao et al., 2024), StepGame (Shi et al., 2022) and LogicAsker (Wan et al., 2024). Additionally, text classification tasks, including Yelp (Yelp Dataset) and DBpedia (Lehmann et al., 2015), as well as the SNLI dataset (Bowman et al., 2015), are included in the comparison. The details of all these tasks could refer to Appendix D.4.

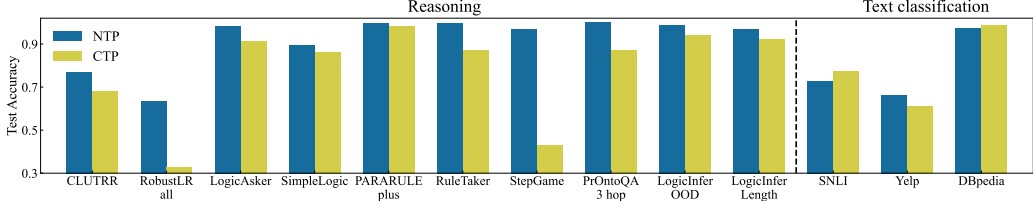

Figure 5: Performance comparison of NTP and CTP across various reasoning tasks. NTP consistently outperforms CTP in reasoning tasks, while performance on text classification tasks is more mixed. All the tasks are trained on the GPT-2 model (125M) from scratch to dismiss the effect of NTP in the pretraining stage. The accuracy is reported when the learning process becomes stable.

The following findings are systematically presented and analyzed in Fig. 5, which provides a comprehensive comparison of both approaches across different task categories. Our experimental results demonstrate that NTP exhibits superior performance compared to CTP across various reasoning-intensive tasks, including PrOntoQA, LogicAsker, and Ruletaker. Particularly noteworthy is NTP's exceptional capability in handling the challenging RobustLR task, where it partially captures underlying logical patterns, while CTP remains stagnant at random-guess levels. As evidenced in Appendix D.4, NTP demonstrates accelerated learning speed for tasks requiring strong reasoning capabilities. However, in text classification tasks that demand less sophisticated reasoning, the performance disparity between NTP and CTP diminishes significantly. In these scenarios, CTP exhibits a slight advantage in learning efficiency, as demonstrated by its comparable performance on SNLI and marginally better convergence rate on the DBpedia dataset.

## 6 ROBUSTNESS OF NEXT TOKEN PREDICTION

In this section, we investigate the robustness of NTP-trained models at both the input and parameter levels. Given that training procedures similar to CTP have been shown to negatively impact robustness (Wang et al., 2023), a specific analysis of NTP is warranted. We also explore the relationship between flatness and NTP's generalization ability empirically. Due to space constraints, we defer to the Appendix A.5 a detailed analysis of model behavior in the presence of erroneous training data. Briefly, we find that NTP models exhibit greater resilience to such errors, whereas CTP exhibits the opposite behavior.

### 6.1 EMBEDDING NOISE

The most straightforward approach to evaluating model robustness involves introducing controlled noise perturbations to the input data and quantitatively measuring the corresponding degradation in model accuracy. Following the settings applied in NETFune (Jain et al., 2023), noise restricted by the sequence length and model hidden size is added after the embedding layer as follows:

$$\text{emb} \leftarrow \text{emb} + \frac{\alpha}{\sqrt{Sd}}\epsilon, \tag{9}$$

where the noise $\epsilon$ is uniformly sampled from the range $[-1, 1]$, and $S, d$ represent for sequence length and embedding dimension separately.

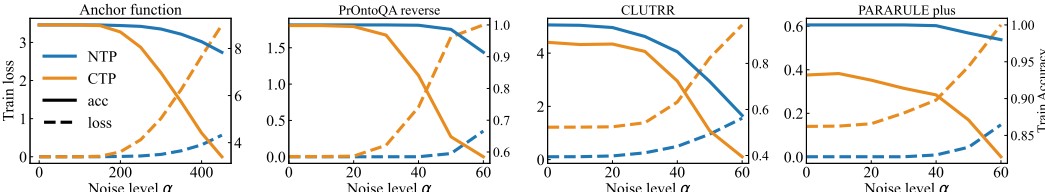

Figure 6: Effect of embedding noise on model performance in different reasoning tasks. The x-axis represents the perturbation strength $\alpha$ in Eq. equation 9 while the y-axis represents the influenced loss (left axis) and accuracy (right axis). NTP-trained models maintain higher accuracy under varying levels of input noise compared to CTP-trained models, which suffer from significant performance degradation in both accuracy and loss.

We have done a thorough analysis of the anchor function, as shown in Fig. 6(a), models trained with NTP are more stable under noise, while CTP-trained models exhibit high sensitivity. In contrast to CTP, the NTP helps the model maintain its learned reasoning solution not only on the embedding layer, but also on the output of different transformer blocks. With Fig. 6, on highly inference tasks like PrOntoQA and PARARULE plus, the performance patterns of NTP and CTP demonstrate remarkable similarity to their performance on the anchor function.

## 6.2 NTP REACHES FLATTER MINIMA

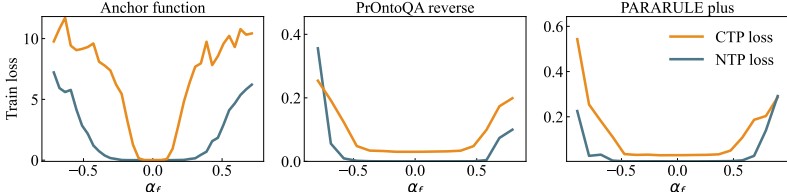

Figure 7: The flatness for task anchor function, PrOntoQA reverse, and PARARULE plus tasks. To alleviate the computational cost, the flatness is calculated on randomly sampled 20,000 instances from the training set.

Flatness, a concept introduced in (Hochreiter & Schmidhuber, 1997) and applied to neural networks in (Keskar et al., 2017), is commonly used to understand model generalization ability. The random direction method (Li et al., 2018) is a widely used approach to assess model flatness or robustness. This method perturbs the model parameters by adding a random vector scaled by the model's norm and a perturbation intensity $\alpha_f$. Let $\theta_N$ and $\theta_C$ denote the parameters of the NTP and CTP trained models, and let $v$ be a random direction in the parameter space. The perturbed parameters are given by:

$$\theta'_N = \theta_N + \alpha_f \frac{\|\theta_N\|}{\|v\|} v, \qquad \theta'_C = \theta_C + \alpha_f \frac{\|\theta_C\|}{\|v\|} v. \tag{10}$$

We tested the performance of NTP and CTP models under a moderate $\alpha_f$, which is shown in Fig. 7. For the simple anchor function task, the solution of NTP is flatter than that of CTP significantly. The flatness disparity between the two training approaches becomes smaller in other tasks since in these tasks the model only needs to discriminate between *true* and *false* responses.

To further investigate the regularization dynamics, we monitored the evolution of the Hessian matrix and parameter norms using the anchor function task. As illustrated in Fig. 8, NTP exhibits a strong regularization effect by significantly suppressing the maximum eigenvalue of the Hessian. This drives the model towards flatter regions of the loss landscape. In contrast, CTP shows a continuous increase in the maximum eigenvalue. This observation suggests that in the absence of regularization, the network tends to converge to sharper solutions. Consequently, while CTP remains trapped in sharp minima near the initialization, NTP successfully moves away from the starting point to locate flatter minima.

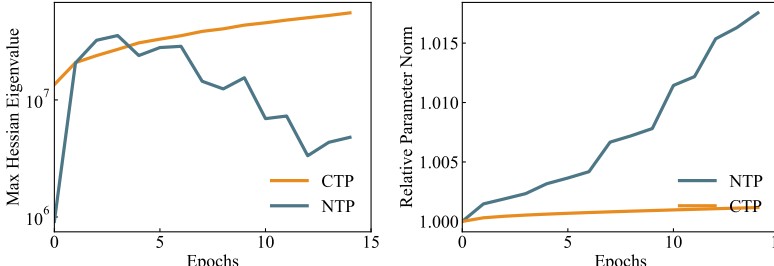

Figure 8: Evolution of the Hessian maximum eigenvalue and parameter norms during training. NTP effectively reduces the maximum eigenvalue, indicating a flatter loss landscape, whereas CTP tends to converge to sharper minima.

## 7 DISCUSSION

**Conclusion** This work systematically investigates the distinct impacts of next token prediction (NTP) and critical token prediction (CTP) training on reasoning tasks, revealing an inherent regularization in NTP-trained models as a key finding. We propose that implicit noise in the training data induces an emergent regularization effect functionally analogous to explicit $L^2$ regularization. This hypothesis is rigorously validated theoretically and empirically through our proposed anchor function composition task. Crucially, empirical validation across realistic tasks confirms that this regularization persists in practical settings, mirroring the insights gained from the composition tasks. Finally, our analysis further investigates the robustness and flatness of models trained with NTP and CTP, demonstrating additional benefits of NTP training.

**Fairness of Comparison between NTP and CTP** We further address the fairness of comparing NTP and CTP, particularly concerning the total training token assumption (for backpropagation). With increased training data or training epochs allocated to CTP, our results show that the NTP-trained model still achieves superior performance on the anchor function Task 2 and the star graph task. This outcome is notable, especially considering that the star graph task has been shown to be inherently challenging or potentially impossible for NTP to train perfectly (Bachmann & Nagarajan, 2024). The experimental evidence is presented in Fig. 9, and further details can be found in Appendix A.2. In real-world scenarios, the significant length disparity between prompt and answer tokens complicates a direct comparison. However, it is worth noting that CTP effectively holds the advantage in the training setting: within the NTP objective, the loss contribution from the critical token is heavily diluted by the numerous context tokens, whereas CTP focuses exclusively on the target.

## LLM USAGE

In this work, the LLMs are employed to correct grammatical errors and inappropriate words.

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

# A    FURTHER DISCUSSION

## A.1    MORE EXPLANATIONS ON NTP SHUFFLE EXPERIMENT IN SEC. 5.1

One might be concerned that the reason for NTP outperforming CTP is simply that CTP does not process question tokens and thus fails to learn the QA dependencies properly. As we pointed out in Sec. 5.1, while semantic information can boost NTP training, it is not the essential factor driving the regularization.

To supplement the experimental details presented in Sec. 5.1, we highlight that the NTP loss function was modified from its original form, typically written as $\widetilde{\mathcal{L}_N} = -\frac{1}{T}\sum_{t=1}^{T-1} 1(x_{t+1})\log(P(x_{t+1}|x_{\leq t}))$, to $\mathcal{L}_N = -\frac{1}{T}\sum_{t=1}^{T-1} 1(\widetilde{x_{t+1}})\log(P(x_{t+1}|x_{\leq t}))$, where $\tilde{x}$ represents a random shuffle of the original sequence $x$. This adjustment was specifically designed to disrupt the natural semantic continuity of the sequence, thereby preventing semantic information from aiding NTP in learning the task dependencies and ensuring that the experiment isolates the effect of the prediction objective itself. The corresponding change in code implementation was from

Listing 1: Before shuffle

```
1  shift_inputs = inputs[:-1]
2  outputs = model(shift_inputs)
3  shift_labels = inputs[1:]
4  NTP_loss = CE(outputs, shift_labels)
```

to

Listing 2: After shuffle

```
1  shift_inputs = inputs[:-1]
2  outputs = model(shift_inputs)
3  shift_labels = inputs[1:-1]
4  answer = inputs[-1]
5  random.shuffle(shift_labels)
6  shift_labels = shift_labels + answer
7  NTP_loss = CE(outputs, shift_labels)
```

## A.2    DETAILS ON FAIR COMPARISON OF TOTAL TOKEN CONSUMPTION BETWEEN NTP AND CTP

This section provides a detailed explanation of the comparison fairness experiments briefly discussed in Sec. 7. The anchor function task and star graph task were chosen for this comparison due to their suitability, characterized by short sequence lengths and the absence of semantic interference (e.g., the anchor function sequence has a fixed length of 9, in contrast to PrOntoQA where the length is around 300 and variable).

**Equalizing Token Consumption via Increased CTP Training Data Volume**    As mentioned in Sec. 7, to ensure fair token consumption between NTP and CTP, we conducted experiments on anchor function Task 2 where CTP's training dataset volume was scaled to more than $9 \times$ that of NTP, thereby matching the total token usage. The results are presented in Fig. 9, with the x-axis indicating the volume of training data used for CTP relative to NTP. For completeness, results from the NTP-shuffle experiment and a PrefixLM (GPT) baseline are also included. As shown in Fig. 9, increasing CTP's data volume resulted in only a slight improvement in its ability to learn inferential solutions, and it consistently underperformed NTP.

**Equalizing Token Consumption via Extended CTP Training Epochs**    For the star graph task, utilizing the publicly available dataset, we extended CTP's training time until its token usage matched that of NTP. The results, presented in Fig. 10, demonstrate that NTP consistently out-performed CTP on the star graph $G_{3,3}$ task under these equalized token conditions (edge list 50, reverse set to False).

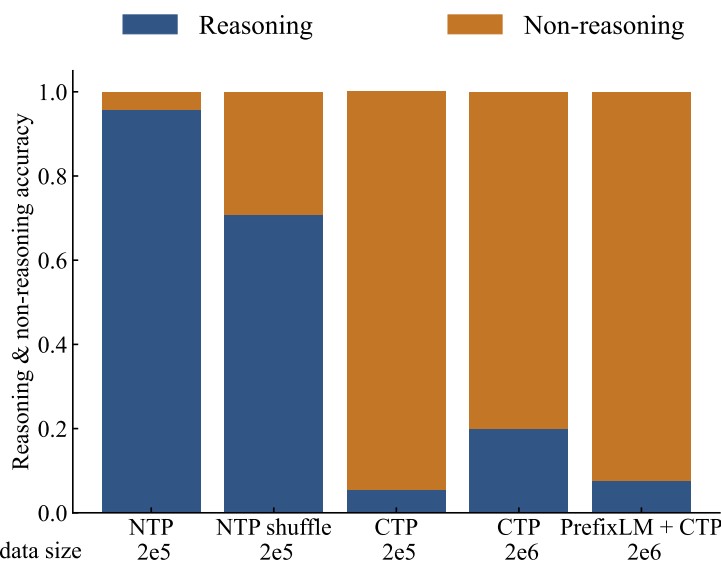

Figure 9: Column 1: NTP trained on 200,000 samples. Column 4: CTP trained on 2,000,000 samples (to match NTP's token usage). Additional experiments: Column 2: NTP with shuffled question tokens (as in PrOntoQA). Column 5: PrefixLM (GPT) baseline.

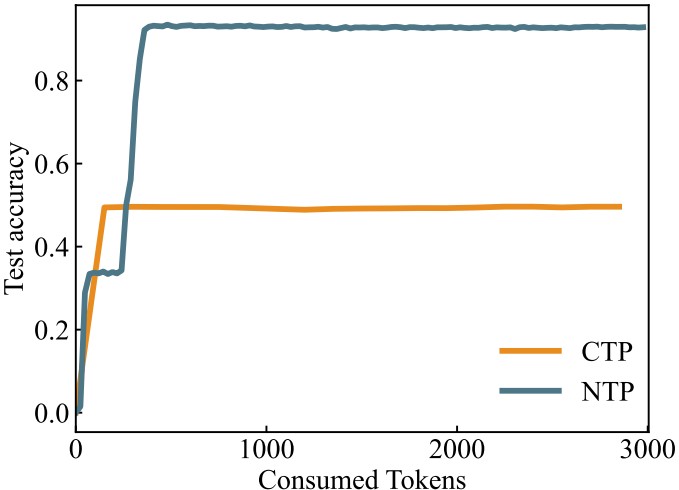

Figure 10: Star graph $G_{3,3}$ task, which is detailed in (Bachmann & Nagarajan, 2024; Setlur et al., 2024), where it was shown that NTP is unable to complete the task, yet the training performance of NTP remains superior to that of direct CTP.

## A.3 RELATIONS BETWEEN CTP AND SFT

When adapting pretrained models for the downstream tasks, CTP (or SFT) is typically preferred over NTP. We evaluated the performance of NTP and CTP on the PrOntoQA dataset using a pretrained GPT-2 model. The results, depicted in Fig. 11(a), show that CTP significantly outperforms NTP in terms of learning speed. This can be attributed to two factors: first, the pretrained model initialized through NTP already resides in a region of the loss landscape that is more amenable to generalization; second, pretraining endows the model with a certain level of reasoning capabil-

ity. Consequently, additional noise in the corpus is unnecessary for aiding generalization, and the absence of noise allows the network to learn the mapping relationships more rapidly.

To our knowledge, the NTP loss function incorporates a component from CTP. We can isolate the CTP portion within the NTP loss and refer to the remaining part as the "noise loss". Subsequently, we experimented with pretraining on both the anchor function task and PrOntoQA using this noise loss, followed by continued training with the CTP loss. This approach demonstrated improved generalization capability compared to directly training with CTP, evidenced by the reasoning solution obtained in the test accuracy reaching 100% on PrOntoQA.

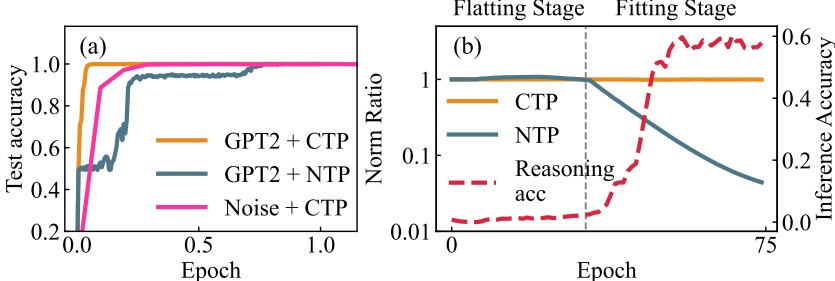

Figure 11: (a) The original 2-hop PrOntoQA task trained on the pretrained models. The legend entry *GPT-2* denotes the pretrained GPT-2 model parameters, and the *Noise* denotes the model is pretrained on the noise term in PrOntoQA by NTP. (b) The ratio of gradient norm on random token position $t = T - 1$ and critical token position $t = T$ of output in Eq. equation 2. The flatting stage and fitting stage are annotated, which corresponds with the reasoning accuracy raise.

### A.4 WHY MODELS ARE NOT MISLED BY NOISE

Inspired by research on the performance of the BERT pretrained model with noisy data (Tänzer et al., 2022), we noticed some phenomena associated with why NTP-trained models are not misled by noise terms. Using the clean anchor function as example, we discover that the gradient norm of the noise terms significantly decreased compared to the critical token (in Fig. 11 (b)), indicating that the network temporarily shifts its focus away from the noise during the learning process. The NTP learning process could be decomposed into two distinguishable stages: The flatting stage, the transformer trying to learn the distribution of the whole sequence. Fitting stage, after the $\mathcal{L}_{noise}$ reaches the lower bound entropy loss of NTP, the transformer notices the regularity of 'key' item and gradually it turns to reasoning solution.

### A.5 MORE EVIDENCE ON THE ROBUSTNESS OF NTP

Another prevalent methodology for robustness evaluation involves deliberately introducing a proportion of noised samples into the training set, subsequently assessing the model's resilience to poisoned data. The addition is a tiny inference task widely used as a test set in the construction of new reasoning techniques of LLM (Deng et al., 2024) and is the basic part of math reasoning steps (Ying et al., 2024).

Our addition dataset consists of addition problems bounded by 1000 and includes several random tokens corresponding to the random $x_i$ in the anchor function. When the length of the random token sequence is $n$, we denote it as Addition-R$n$. The numbers are padded to 4 digits and split into individual digits by the tokenizer. The total number of samples is $D = [0, 1000]^2$.

In the error addition task, we remove a square region from the center of $D$ with side length 100, denoted as $H = [400, 600]^2$. We randomly select 1000 or 2000 samples in $H$ and add noise $\pm 50$ to the labels, denoted them as poisoned samples $D_e$. The training set consists of $D \backslash H \cup D_e$, which includes the error samples, and the test dataset is $H \backslash D_e$. Drawing insights from our experience with anchor functions, we utilize an 8-layer transformer and observe the influence of poisoned samples.

Fig. 12 shows both NTP and CTP could easily learn addition without any poisoned samples. When the noise is introduced in the training data, NTP demonstrates superior performance, as evidenced

by its higher peak test accuracy and delayed accuracy degradation compared to CTP. From Fig. 12 (b), meanwhile, CTP is trapped in memorizing poisoned samples at a faster speed than NTP.

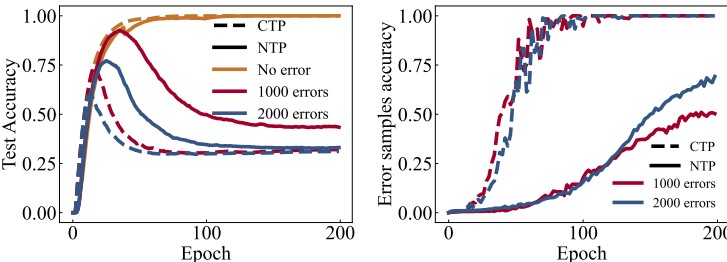

Figure 12: Comparison of NTP and CTP on the addition task with varying poisoned samples. The *1000 errors* and *2000 errors* denote training scenarios where an 800,000-sample dataset was deliberately contaminated with precisely 1,000 or 2,000 erroneous data points, respectively. (a) Test accuracy (on $H \backslash D_e$) (b) the memorizing speed of the poisoned samples $D_e$. The CTP could easily fit the errors before 100 epochs whereas NTP fits at a lower speed.

## B  THEORETICAL PROOFS

We follow the definition of Fisher information matrix in (Liu et al., 2023) and restate it as:

**Definition 2.** *For a parameterized random variable X, let $p(X; \boldsymbol{\theta})$ be the probability density function for X. Then the fisher information matrix has typical element*

$$[\mathcal{I}(\boldsymbol{\theta})]_{i,j} = \mathbb{E}\left[\left(\frac{\partial}{\partial \theta_i} \log p(X; \boldsymbol{\theta})\right)\left(\frac{\partial}{\partial \theta_j} \log p(X; \boldsymbol{\theta})\right)\right]$$

### B.1  PROOF OF THEOREM. 1

For the convenience of notation we set $\mathcal{L}_C \leftarrow \frac{1}{T}\mathcal{L}_C$ in the following proof.

*Proof.* By definition of NTP and CTP loss, the following equation holds after proper normalization:

$$\mathcal{L}_N = \mathcal{L}_C - \frac{1}{NT}\sum_{i=1}^{n}\sum_{t=1}^{T} 1_{\{x_{t+1}^i\}} \log P_{\boldsymbol{\theta}}(x_{t+1}^i | x_{\leq t}^i),$$

where each token is where each token is uniformly sampled from the vocabulary set $\mathcal{V}$. The indicator function can be equivalently expressed using Dirac delta notation for the second term:

$$-\frac{1}{NT}\sum_{i=1}^{N}\sum_{t=1}^{T} 1_{\{x_{t+1}^i\}} \log P_{\boldsymbol{\theta}}(x_{t+1}^i | x_{\leq t}^i) = -\frac{1}{NT}\sum_{i=1}^{N}\sum_{t=1}^{T}\sum_{s=1}^{V} \delta(s, x_{t+1}^i) \log P_{\boldsymbol{\theta}}(s | x_{\leq t}^i)$$

Since each token is uniformly sampled, the second term could be decomposed into two parts:

$$-\frac{1}{NT}\sum_{i=1}^{N}\sum_{t=1}^{T}\sum_{s=1}^{V} \delta(s, x_{t+1}^i) \log P_{\boldsymbol{\theta}}(s | x_{\leq t}^i) = -\frac{1}{NT}\sum_{i=1}^{N}\sum_{t=1}^{T}\sum_{s=1}^{V} \frac{1}{V} \log P_{\boldsymbol{\theta}}(s | x_{\leq t}^i)$$

$$-\frac{1}{NT}\sum_{i=1}^{N}\sum_{t=1}^{T}\sum_{s=1}^{V}(\delta(s, x_{t+1}^i) - \frac{1}{V}) \log P_{\boldsymbol{\theta}}(s | x_{\leq t}^i)$$

By discarding constants independent of the parameters $\boldsymbol{\theta}$, we have

$$-\frac{1}{NT}\sum_{i=1}^{N}\sum_{t=1}^{T}\sum_{s=1}^{V} \delta(s, x_{t+1}^i) \log P_{\boldsymbol{\theta}}(s | x_{\leq t}^i) = \frac{1}{NT}\sum_{i=1}^{N}\sum_{t=1}^{T} \text{KL}(\text{Unif}(\mathcal{V}), P_{\boldsymbol{\theta}}(\cdot | x_{\leq t}^i))$$

$$-\frac{1}{NT}\sum_{i=1}^{N}\sum_{t=1}^{T}\sum_{s=1}^{V}(\delta(s, x_{t+1}^i) - \frac{1}{V}) \log P_{\boldsymbol{\theta}}(s | x_{\leq t}^i).$$

Based on the assumption, we have $\text{Unif}(\mathcal{V}) = P_{\boldsymbol{\theta}_0}(s|x^i_{\leq t})$ for all $i, t$. So the first term can be rewritten as

$$\frac{1}{NT} \sum_{i=1}^{N} \sum_{t=1}^{T} \text{KL}(\text{Unif}(\mathcal{V}), P_{\boldsymbol{\theta}}(\cdot|x^i_{\leq t})) = \frac{1}{NT} \sum_{i=1}^{N} \sum_{t=1}^{T} \text{KL}(P_{\boldsymbol{\theta}_0}(\cdot|x^i_{\leq t}), P_{\boldsymbol{\theta}}(\cdot|x^i_{\leq t}))$$

By expanding the KL divergence , we derive the implicit regularization term

$$\frac{1}{NT} \sum_{i=1}^{n} \sum_{t=1}^{T} \text{KL}(P_{\boldsymbol{\theta}_0}(s|x^i_{\leq t}), P_{\boldsymbol{\theta}}(s|x^i_{\leq t})) = \frac{1}{NT} \sum_{i=1}^{n} \sum_{t=1}^{T} \frac{1}{2}(\boldsymbol{\theta} - \boldsymbol{\theta}_0)^{\mathsf{T}} I_{\boldsymbol{\theta}_0}(x^i_{\leq t})(\boldsymbol{\theta} - \boldsymbol{\theta}_0) + \mathcal{O}(\|\boldsymbol{\theta}\|^3).$$

(11)

To control for the residual term, we change the order of summation.

$$-\frac{1}{NT} \sum_{i=1}^{N} \sum_{t=1}^{T} \sum_{s=1}^{V} (\delta(s, x^i_{t+1}) - \frac{1}{V}) \log P_{\boldsymbol{\theta}}(s|x^i_{\leq t}) = -\frac{1}{NT} \sum_{t=1}^{T} \sum_{s=1}^{V} \sum_{i=1}^{N} (\delta(s, x^i_{t+1}) - \frac{1}{V}) \log P_{\boldsymbol{\theta}}(s|x^i_{\leq t}).$$

Using Chebyshev's Inequality, there exists $C$ independent of parameters such that

$$\mathbb{P}\left( \cup_{s=1}^{V} \left| \frac{1}{N} \sum_{i=1}^{N} (\delta(s, x^i_{t+1}) - \frac{1}{V}) \right| > \delta \right) \leq \frac{C}{\delta^2 N}.$$

As a result, for any $\varepsilon > 0$, let $\frac{C}{\delta^2 N} = \varepsilon$, we obtain $\delta = \frac{C}{\sqrt{\varepsilon N}}$. Then, the asymptotic probabilistic bound holds

$$\mathbb{P}\left( \cup_{s=1}^{V} \left| \frac{1}{N} \sum_{i=1}^{N} (\delta(s, x^i_{t+1}) - \frac{1}{V}) \right| > \frac{C}{\sqrt{\varepsilon N}} \right) \leq \varepsilon$$

As a result, we finish the proof by

$$\left| \frac{1}{NT} \sum_{t=1}^{T} \sum_{s=1}^{V} \sum_{i=1}^{n} (\delta(s, x^i_{t+1}) - \frac{1}{V}) \log P_{\boldsymbol{\theta}}(s|x^i_{\leq t}) \right| = \mathcal{O}(\frac{1}{\sqrt{\varepsilon N}})$$

$\square$

## B.2 PROOF OF PROPOSITION 1

In this section, we focus on the empirical fisher information matrix: $\frac{1}{T} \sum_{t=1}^{T-2} \mathbb{E}_{X_t} I_{\boldsymbol{\theta}_0}(X)$. For the models defined like Def. 1, we only need to consider $\nabla_{W_{\text{proj}}} \log P_{\boldsymbol{\theta}}$ because $\nabla_{\hat{\boldsymbol{\theta}}} P_{\boldsymbol{\theta}} = 0$ at origin. So we take $\hat{\boldsymbol{\theta}} = 0$ and the logit is

$$\sum_{t'=1}^{t} \lambda_{t,t'} x_{t'} W_{\text{proj}}.$$

(12)

For logit vector $\ell$, we have

$$\frac{\partial \log P_s}{\partial \ell_k} = \delta_{ks} - P_k.$$

Then, we get

$$\nabla_{W_{\text{proj}}} \log P_s = \sum_{k=1}^{d_{\text{vob}}} (\delta_{ks} - P_k) \left( x_t + \frac{1}{t} \sum_{t'}^{t} x_{t'} \right) e_k^{\mathsf{T}}$$

$$= \left( \sum_{t'=1}^{t} \lambda_{t,t'} x_{t'} \right) \left( e_s - \frac{1}{d_{\text{vob}}} \mathbf{1} \right)^{\mathsf{T}}$$

Let $u_t := \sum_{t'=1}^{t} \lambda_{t,t'} x_{t'}$ and $v_s := e_s - \frac{1}{d_{\text{vob}}} \mathbf{1}$, we get $\nabla_{W_{\text{proj}}} \log P_s = u_t v_s$. Using the identity $\text{vec}(uv^{\mathsf{T}}) = v \otimes u$, we have $\text{vec}(\nabla_{W_{\text{proj}}} \log P_s) = v_s \otimes u_t$. Then,

$$I_0(x^i_{\leq t}) = \frac{1}{d_{\text{vob}}} \sum_{s=1}^{d_{\text{vob}}} (v_s \otimes u_t)(v_s \otimes u_t)^{\mathsf{T}}$$

(13)

$$= \frac{1}{d_{\text{vob}}} \sum_{s=1}^{d_{\text{vob}}} (v_s v_s^{\mathsf{T}}) \otimes (u_t u_t^{\mathsf{T}}),$$

where the term $\frac{1}{d_{\text{vob}}}$ comes from the uniform distribution. Since $u_t$ is independent of $s$, we take the sum for $s$ first and get

$$I_0(x^i_{\leq t}) = \frac{1}{d_{\text{vob}}} \left( I - \frac{1}{d_{\text{vob}}} \mathbf{1}\mathbf{1}^T \right) \otimes (u_t u_t^\intercal). \tag{14}$$

Finally, due the the uniformity of noise, there exists $\lambda > 0$ such that the empirical fisher information matrix can be approximated by

$$\frac{\lambda}{d_{\text{vob}}} \left( I - \frac{1}{d_{\text{vob}}} \mathbf{1}\mathbf{1}^T \right) \otimes I, \tag{15}$$

whish finish the proof.

## C  NTP ENHANCES EARLY TRANSFER GENERALIZATION

When the available data for a specific task is insufficient for training a model from scratch, transfer learning typically serves as an effective solution by finetuning a pretrained model with existing knowledge. In this section, we conduct transfer learning experiments between models trained using NTP and CTP across diverse downstream tasks. Our investigation yielded two results: (1) Models trained with NTP demonstrate accelerated generalization during the early stages of finetuning, although both approaches ultimately converge to comparable accuracy levels; (2) NTP-trained models exhibit a higher propensity for catastrophic forgetting during the finetuning process.

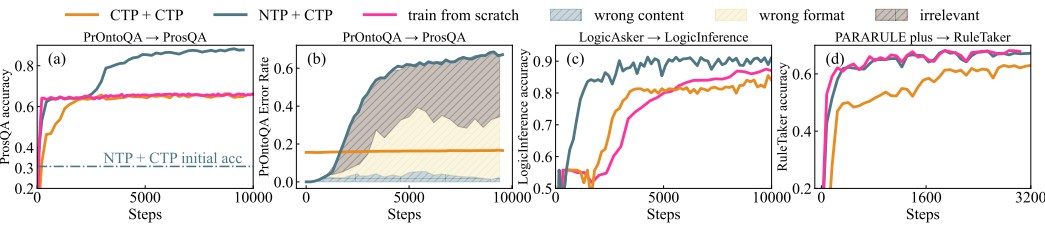

Figure 13: Finetuning results with multiple tasks. *NTP+CTP* means the model is NTP-trained on previous task and CTP-finetuned on post task; *CTP+CTP* means the model is CTP-trained on previous task and CTP-finetuned on post task. *train from scratch* means the model is trained from scratch with the same configuration of CTP-funtuning. (a, b) 2-hop PrOntoQA models continue to train on ProsQA. (a) The accuracy of ProsQA test data with the CTP finetuning process. (b) The accuracy of PrOntoQA test data and and the proportion of three error types of *NTP+CTP* during finetuning. The *wrong content*, *wrong format* and *irrelevant* represent incorrect answer content, improper answer formatting, and irrelevant responses. Regarding the omitted *CTP+CTP* error types visualization, its *wrong content* metric consistently maintains at 1.0, which demonstrates its immunity to finetuning perturbations. (c, d) More examples of transfer learning capability difference between NTP and CTP.

The ProsQA dataset, proposed in (Hao et al., 2024), represents an enhanced version of PrOntoQA, featuring more explicit reasoning graph structures. However, its limited scale precludes its use for training models from scratch. In this section, we primarily leverage its advantage of providing answer contrastive pairs to conduct finetuning experiments on models initially trained using both NTP and LTP on the 2-hop original PrOntoQA dataset.

We employed a relatively low learning rate (2e-6) to meticulously capture the accuracy transitions between the original PrOntoQA 2-hop task and the new ProsQA task. The experimental results in Fig. 13 (a) demonstrate that the NTP model successfully predicts a portion of the validation set at the beginning, consistently outperforming CTP throughout the training process. This empirical evidence strongly suggests that NTP-trained models have inherent advantages for transfer learning applications.

However, in Fig. 13 (b), our empirical findings indicate that NTP-trained models are potentially more susceptible to catastrophic forgetting compared to their CTP counterparts. Through systematic evaluation, we observed a pronounced accuracy degradation on the original PrOntoQA dataset for

NTP models as finetuning progressed, whereas CTP models showed only marginal performance decline, consistently maintaining a superior accuracy level.

Furthermore, We conducted an in-depth analysis of prediction errors, categorizing them into three distinct types: (1) Wrong content: instances where the model incorrectly predicts 'False' when the ground truth is 'True'; (2) Wrong format: cases such as responding with 'shy' instead of the required 'True/False' format to the question "Is Rex shy?"; and (3) Irrelevant responses: The responses contains unrelated words from the input sentence. Our finding suggests the NTP-trained models are more willing to transfer the answer from PrOntoQA into new formats, ProsQA, while CTP-trained models demonstrate more consistent performance on PrOntoQA, even when the ProsQA task semantics remain identical. It treats the tasks separately and, as a consequence, shows weaker transfer ability.

Given the scale limitations of the dataset, we conducted additional experiments with multiple data groups to evaluate the transfer capabilities of NTP and CTP. Across various experimental settings, NTP consistently demonstrated superior transfer characteristics, even when the tasks were not directly related but shared similar reasoning patterns, as Fig. 13 (c, d) shows.

## D  EXPERIMENTAL FRAMEWORK AND IMPLEMENTATION DETAILS

This section provides a detailed description on the experimental implementations.

### D.1  MORE EXPLANATION ON PROBLEM SETUP OF ANCHOR FUNCTION

**Model architecture**    For self-attention block $\text{Attn}$ we have

$$\text{Attn}(X) = \text{softmax}\left(\frac{XW_QW_K^TX^T}{\sqrt{d_k}}\right)XW_V. \tag{16}$$

And the fully connected block is

$$\text{MLP}(X) = \text{ReLU}(XW_1)W_2. \tag{17}$$

For realistic reasoning tasks, we initialize the weight with zero-mean normal distribution with a standard deviation of 0.02 default by Hugging Face.

**Data Generation**    Since we have fixed the anchor set $\mathcal{A}$, then for composition task shown in Eq. equation 4, 16 anchor pairs exist in total. We generate 900,000 samples in total and partition it into training and testing subsets with a 9:1 ratio. Each anchor pair $(a, b)$ shares the equal number of samples. Then we generate the dataset $X$: The position of anchor and key are randomly selected in the fixed-length sequence, and the other positions are filled with random number from $\mathcal{D}$. The last token is replaced by the function solution of the sequence, i.e.

$$X = \{x_i \in \mathcal{D}, a, b \in \mathcal{A} \big| [x_1, \ldots, x_i, a, b, \ldots, x_n, (a, b)(x_i)]\}. \tag{18}$$

For anchor function and realistic tasks, we use vallina GPT-2 model with 12 layers and 12 heads, embedding dimension is set as 768. We forbid dropout in the residual, embedding and attention branch, to avoid effect of regularization methods. We set the learning rate is 5e-5 with linear warmup scheduler (warmup ratio = 0.1). The batch size of anchor function is 2000 and for realistic tasks, is set to 160. Without any mention, the weight decay is set as 0 and we use seed 42 by default.

### D.2  DETAILS ON REALISTIC TASKS

To construct the dataset for CTP training, we equipped the answer with a separation mark, use the PrOntoQA for example, we turn the sequence into:

> Gwompuses are zhorpuses. Every chorpus is transparent. Each gerpus is a boom-pus. Bompuses are sweet. Each bompus is a felpus. Bompuses are yerpuses. Felpuses are not fast. Each felpus is a terpus. Each timpus is fast. Felpuses are quimpuses. Every zhorpus is brown. Every kerpus is earthy. Kerpuses are rorpuses. Fae is a felpus. Fae is a kerpus. Question: True or false: Fae is fast. **[SEP]** False **[SEP]**

Like SFT, the loss of CTP is only calculated tokens between the **[SEP]** symbols.

### D.3 ADDITION TASK

The addition task is designed to show the robustness of NTP training. We borrow the reverse addition settings, like $314 + 518 = 832$, changed into $413 + 815 = 238$. Given the pure addition doesn't contain any noise in the corpus, We intentionally introduce noise tokens into the dataset. The reconstructed sequence is, for example,

7, 9, 1, 1, [SEP_R], 5, 5, 4, 0, +, 3, 5, 4, 0, [SEP_R], 4, **[SEP]** 8, 0, 9, 0 **[SEP]**

The symbol [SEP_R] is used to remind the model of the start and end in equation, and the first **[SEP]** could be regarded as the equal symbol '='. Outside the symbol [SEP_R], we add 5 noise terms to help simulate the noise in anchor function. The training configurations follow the anchor function with a 8 layers 4 heads prenorm model.

### D.4 THE REASONING TASK DATASETS OVERVIEW

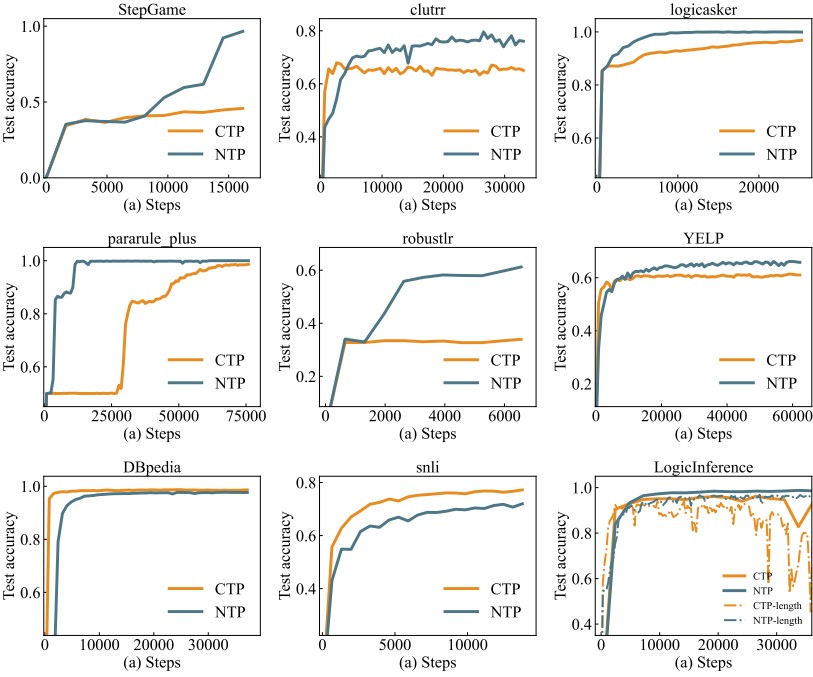

Figure 14: The NTP and CTP training process of reasoning tasks and text classification tasks. CTP outperforms NTP on tasks that involve shorter texts and require less extensive reasoning like DBpedia or SNLI but NTP outperforms CTP on reasoning data, such as PrOntoQA, RobustLR etc. The first figure PrOntoQA-2hop cloze means the accuracy of cloze version about PrOntoQA.

**PrOntoQA & ProsQA**  Every sequence in PrOntoQA dataset consists of three parts: fact, question and answer. Some noise disturbance terms are mixed in the fact part. An example of 1-hop reasoning is below:

> Fact:
> Every gwompus is not amenable. Every gwompus is a chorpus. Gwompuses are zhorpuses. Every chorpus is transparent. Chorpuses are gerpuses. Every chorpus is a storpus. Gerpuses are not hot. Gerpuses are bompuses. Each gerpus is a boompus. Bompuses are sweet. Each bompus is a felpus. Bompuses are yerpuses. Felpuses are not fast. Each felpus is a terpus. Each timpus is fast. Felpuses are

quimpuses. Quimpuses are nervous. Each yerpus is not discordant. Each boompus is sunny. Storpuses are wooden. Every zhorpus is brown. Every kerpus is earthy. Kerpuses are rorpuses. Fae is a felpus. Fae is a kerpus.
Question:
True or false: Fae is fast.
Answer:
False

We could see that the question's answer only depends on the fact, where the inference chain is underlined. So it's possible that the different fact causes the same queries share different answer. In the PrOntoQA reverse dataset, we harmonized the answers to the same questions in the train dataset. In each sequence, the question could be referred to the form 'A is B?'. We define the OOV dataset as the A and B have never appeared in the train dataset. The accuracy on OOV dataset reflects whether the model learned the rule behind PrOntoQA. These could refer to Fig. 2(b).

Two new versions are involved in the paper, ProsQA and PrOntoQA cloze. The cloze-style version transforms the question 'Question: True or false: Fae is fast. Answer: False' into 'Question: Fae is _____ Answer: fast.' The ProsQA version comes from (Hao et al., 2024), prepares a disturbance options on the result:

Question: Fae is fast or shy? Answer: fast.

We used 500,000 samples for training and 5,000 samples for validation or testing with respect to every PrOntoQA experiment (original, cloze, and reverse). We applied all the data in ProsQA, where there are 18,186 samples for train and 500 for test.

**LogicInference** he LogicInference dataset primarily comprises propositional logic problems and a curated subset of first-order logic formulations. We conducted a two-stage filtering process: initially isolating the first-order logic instances, followed by selecting those containing well-formed yes/no question-answer pairs that are particularly suited for CTP.

Fact:
Consider the following premises. exists x15: $R15(x15) \to U1(x15)$. forall x15: $Q15(x15) \to Q10(x15)$. forall x15: $\sim P15(x15)$ or $R15(x15)$. forall x15: $P15(x15)$ or $Q15(x15)$. forall x15: $Q(x15)$. forall x15: $Q10(x15) \to U1(x15)$.
Question:
Can we infer exists x15: $U1(x15)$ and $Q(x15)$ from them?
Answer: yes

**CLUTRR** CLUTRR is a diagnostic benchmark designed to evaluate the robustness of natural language understanding systems. It tasks models with inferring kinship relations from short stories, requiring both relationship extraction and logical rule deduction. Each story features a complete family structure and requires the model to infer the relationships between any two family members.

Facts:
Stella's husband, Albertus, surprised her with tickets to a football game for their anniversary. Albertus rushed to the hospital to find out that his wife had already given birth to a boy and had named him Pleasant. Frank told a secret to her sister, Blanche. Blanche passed it along to her brother, Pleasant. Pleasant took his Aunt Frank out for her favorite meal. Barnett is Frank's older brother. He has never liked any of her boyfriends. Blanche and her aunt, Frank, went to the deli. They got half a pound of corned beef and two pounds of salami. Gina asked her daughter, Frank, if she had fun at school that day. Frank answered that she and her sister, Frank, had lots of fun together. Albertus went to the game with his sister Frank. Albertus took his daughter Gertie to the park that afternoon to play. Pleasant's wife, Celestia, surprised him on his birthday. He couldn't believe she pulled it off. Florence and her son's wife, Celestia, flew first class to see the concert.
Question: Blanche is who of Stella
Answer: daughter

**LogicAsker**    LogicAsker systematically assesses reasoning by employing atomic skills based on propositional and predicate logic. The LogicAsker dataset features relatively low difficulty and contains few distractors. We sampled 500,000 data for train and 12,000 data for test. An example is:

> Statement:
> For all x12, x12 will go running. For all x12, x12 is a police officer. There is at least one x12 for which if x12 were a scientist, then x12 is not a police officer.
> Question:
> Can we infer the following from them? Answer yes or no: There is at least one x12 for which x12 is not a scientist
> Answer: yes

**PARARULE Plus**    PARARULE Plus is a deep multi-step reasoning dataset over natural language based on the closed-world assumption. It is derived from the PARARULE dataset and has deeper samples. Similar with the PrOntoQA dataset, it also consists of facts, question and answer. However, it surpasses PrOntoQA in terms of sentence complexity.

However, there is an implicit unreasonable settings in the original dataset, is that all the queries with the answer 'true' are end up with the format 'A is B?' and the queries with the answer 'false' are end up with the format 'A is not B?' This causes the transformer learns a shortcut, mapping from existence of 'not' in question to the binary answer true or false. From the original settings, both CTP and NTP could easily reach accuracy 1.

We took a deep insight in the generalization rules of PARARULE plus, and rewrote some of them to decouple the answers from the format of queries. We added 4 new rules and redo the same experiments. We use depth-2 dataset for train (500,000 samples) and for test (5,000 samples).

> Fact:
> The wolf is tired. The wolf is dull. The wolf is rough. The wolf needs the dog. The bear sees the rabbit. The bear is fierce. The bear is awful. The dog is kind. The dog is smart. The dog is round. The rabbit is cute. The rabbit is lovely. The rabbit is furry. Kind animals are cute. If something is dull then it visits the dog. If something visits the dog then it is slow. If something is tired and dull then it is rough. If something is cute and lovely then it is adorable. If something is fierce and awful then it is obese. If something is rough then it is lazy. All lazy animals are sleepy. If something is cute then it is lovely. All lovely animals are furry. If something is obese then it is strong. All strong animals are heavy. If something is adorable then it is beautiful. All beautiful animals are small. All slow animals are big.
> Question:
> The bear is not heavy
> Answer: false

**RobustLR**    The authors propose RobustLR for diagnose the robustness to logical variations in language models. Compared to PrOntoQA, this dataset is more comprehensive and specific, while also encompassing a variety of different relations. As a consequence, both NTP and LTP face difficulties learning this problem. The LTP's accuracy is stagnated at the random guessing accuracy. The train and test dataset consist of 210,865 and 8,000 samples separately.

> Statements:
> Fiona is white. Dave is blue. Anne is the uncle of Bob. Charlie is white if Dave is blue. Charlie is white and Dave is not quiet if Fiona is white or Anne is the uncle of Bob. If Fiona is white or Anne is Bob's uncle then Charlie is white and The uncle of Anne is not Gary. If Charlie is white then Anne is big. Bob is nice if Dave is not quiet and Anne is the uncle of Bob. Bob is not nice if Anne is the uncle of Bob and Gary is the mother of Harry. If Dave is blue or Anne is big then Dave is not nice and Bob is nice. If Dave is not quiet and Gary is the mother of Harry then Dave is nice. If Bob is nice or Dave is not nice then Fiona is the aunt of Bob. If Dave is not nice then Bob is not Anne's brother. Bob is Anne's brother if The mother of Harry is Gary. Harry is furry if The brother of Anne is not Bob or

> Gary is not the uncle of Anne. If Charlie is white and Gary is the mother of Harry then Harry is not furry. Anne is not the wife of Dave if Bob is nice and Anne is Bob's uncle.
> Question:
> The mother of Harry is not Gary.
> Answer: True

The statement is confusing and we split it into several parts: Facts, 2-hop Inference and contradiction.

**RuleTaker** The authors developed the RuleTaker dataset through a systematic transformation of natural language into structured reasoning processes, establishing an emulation framework for soft reasoning. For example, we have following sample like:

> Statement:
> Cow sees mouse. Cow likes tiger. Bear is cold. Cow is big. If X visits bald eagle and X is kind then X is nice.
> Question:
> Cow sees bear?
> Answer: False

We use 29,000 samples for training and 1000 for testing.

**SimpleLogic** Aiming to discover the logic capability in BERT models, especially for its OOD generalization ability, the authors constructed the SimpleLogic dataset, with rule-priority and label-priority. We introduce 192,000 training dataset and 1,0000 testing dataset for this task. The example is attached below:

> Assumptions:
> If messy and reserved, then worrisome. If messy and reserved and tender, then weary. If tender, then friendly. If frightened and worrisome, then tender. If reserved, then tender. If weary, then messy. If lonely and weary and tender, then reserved. If tender, then messy. If worrisome and tender and lonely, then messy. If lonely and frightened and friendly, then messy. If reserved and messy and friendly, then worrisome. If reserved, then frightened. If lonely and friendly and messy, then tender. If frightened, then tender. If lonely, then frightened. If lonely, then worrisome. If messy and friendly, then lonely. If weary, then reserved. If reserved and frightened and weary, then tender. If worrisome and reserved and weary, then frightened. If reserved and friendly, then worrisome. If worrisome, then lonely. If messy and worrisome, then lonely. If frightened, then messy. If lonely, then friendly. If weary, then lonely.
> Question: weary worrisome reserved lonely to messy
> Answer: true

**StepGame** StepGame is inspired from bAbI-17/19 benchmarks (Weston et al., 2015) and to mitigate bAbI's limitations, such as fixed expressions, small number of reasoning hops and the lack of noise for robustness test. Each data instance in the dataset describes a set of spatial relationships among multiple objects and requires the model to deduce the relative position between two specified objects based on the given relational information. Similar to PrOntoQA, we generate 500,000 synthetic training dataset and 5,000 testing dataset.

> The object Z is positioned directly above the object K. Object G is above object I and to the right of it, too. N is diagonally to the bottom left of J. A is to the bottom-left of N. K is positioned below and to the right of Y. O is at the lower side of G. Z is to the right of Y. S is placed in the left direction of K. O is directly south east of H. G is to the right of Q. H is placed at the lower right of K.
> Question: What is the relation of the agent O to the agent G?
> Answer: below

**SNLI**  The Stanford Natural Language Inference (SNLI) corpus collects of 570k human-written English sentence pairs for entailment examination. There are 550,152 and 1,000 samples in training and testing dataset. A typical example of SNLI is

> Text: A man inspects the uniform of a figure in some East Asian country.
> Hypothesis: The man is sleeping
> Answer: contradiction

**Yelp**  The Yelp Dataset is a comprehensive collection of data related to reviews of businesses, and is widely used to predicting positive or negative reviews. We use all the 650,000 sequences for training and 50,000 for testing. The format of reviews like:

> Text: To keep it short and sweet: Save yourself $100. Buy a good board game, your alcohol of choice, order a pizza, and invite your friends over. nWhat an incredible disappointment. After seeing the enticing commercials so many times, we decided to give this place a try on a double date. I understand the prices of the play cards and won't dispute them; however, the food was incredibly over-priced, came out COLD (as in, sat on a counter without warmers for a minimum of 30 minutes) and I literally had to ask the bartender if there was any vodka in my drink. It was pure juice. $38 for three shots that had little-no alcohol in them. (Not to mention, my glass was dirty, and I saw the bartender scoop the glass into the ice basin because she was too lazy to use the sanitary scoop. I know the Food and Beverage Commission would be as disappointed as I was.) The service was terrible. Don't ask for anything from your waiter, as they are a little too busy on their cell phones or conversing amongst themselves. Was it fun to be in an adult-themed arcade? Yes. If you're looking for a good atmosphere to go with friends to play games, I suppose I would advise you give it a shot. I would never recommend their food, customer service, or drinks. Save yourself the money and stay home, or go for a traditional bowling, figure skating, roller-blading, rock climbing, basically any other physically-entertaining themed date instead.
> Answer: Negative

**DBpedia**  The DBpedia dataset is designed to evaluate a model's capability to accurately classify news articles into predefined categories based solely on their titles and concise summaries, thereby testing both the model's comprehension of textual semantics and its ability to perform hierarchical classification tasks. The size of training and testing set are 560,000 and 70,000 separately.

> Title: Export-Import Bank of Romania
> Content: Exim Bank is The Export-Import Bank of Romania based in Bucharest.
> Answer: 0

### D.5  EXPERIMENTS COMPUTE RESOURCES

The experiments were conducted on a server with the following configuration:

- 48 AMD EPYC 7352 24-Core Processors, each with 512KB of cache
- 251GB of total system memory
- 8 NVIDIA GeForce RTX 4080 GPUs with 16GB of video memory each
- The experiments were run using Ubuntu 22.04 LTS operating system

