# OpenReview forum: "Uncritical tokens are 'critical' in pretraining: the implicit regularization effect of Next token prediction"
_ICLR.cc/2026/Conference — Submitted to ICLR 2026_

### Official Review · Reviewer_aZhb · 2025-10-17

**Soundness:** 3
**Presentation:** 3
**Contribution:** 2
**Rating:** 4
**Confidence:** 3

**Summary:**

This paper challenges the traditional view that "uncritical tokens" in NTP are training noise, revealing their critical role as implicit regularizers for reasoning tasks. By contrasting NTP with CTP which only optimizes for key tokens like answers, the authors theoretically prove that NTP’s uncritical token loss acts as an L²-like implicit regularizer via Fisher information matrix analysis, and empirically validate that NTP-trained models exhibit superior reasoning ability, generalization, and robustness across synthetic and real-world reasoning benchmarks.

**Strengths:**

* **Important insights**: The paper's central finding—that predicting seemingly irrelevant tokens is beneficial for learning robust reasoning—is counterintuitive and significant.

* **Theoretical and Empirical Synergy**: The work is distinguished by its tight integration of theoretical analysis and empirical validation.

* **Clear Mechanistic Explanations**: Beyond accuracy metrics, the paper uses information flow analysis (Fig. 3b,c) to provide a clear, interpretable visualization of how NTP and CTP lead to different internal reasoning strategies.

**Weaknesses:**

* **Theoretical scope**: The theoretical claims rely on strong assumptions such as vanishing initialization and uniform Fisher information. While these conditions are definitely useful for creating a tractable analytical model, their applicability to real-world, large-scale training scenarios is not fully established.

* **Scalability**: The experiments on GPT-2 inevitably raise concerns about the robustness and generalizability of the conclusions in the current research context, as works such as RHO-1 have demonstrated effectiveness on models ranging from 1B to 7B parameters at a minimum.

* **Clarity**: The paper uses the term "pre-training" in title and main corpus to describe training models from scratch on task-specific datasets. This could be slightly confusing, as "pre-training" typically refers to unsupervised learning on massive, general-domain corpora. Sharpening this terminology to distinguish between "from-scratch task training" and "fine-tuning" would improve clarity.

* **Reasoning gaps**: I understand that the paper’s task setup is carefully designed and helpful for clarification, but I still worry whether there is a genuine need—and whether significant experimental differences would emerge—when applying the NTP and CTP training paradigms to truly challenging mathematical problems, such as real questions from AIME or OlympiadBench.

**Questions:**

* In fig. 3(a), your theoretical analysis compellingly links NTP to L2 regularization under specific initialization conditions. How does this implicit regularization from NTP interact with explicit regularizers like weight decay or dropout?

* Appendix A.3 notes that CTP is more efficient when fine-tuning an existing pre-trained model. Could you expand on the practical implications of this? Does it suggest a hybrid strategy where NTP is ideal for initial pre-training, while CTP (i.e., standard SFT) is optimal for downstream adaptation?

* The robustness experiments demonstrate NTP's resilience to input noise and label errors. Do you believe this is solely due to the flatter minima it finds, or are there other mechanisms at play, such as learning a more distributed or compositional representation of knowledge that is inherently more robust?

* Do NTP’s advantages persist in larger models (1B+)? If not, what is the threshold where implicit regularization becomes redundant?

* Can you quantify the implicit regularization strength of NTP (e.g., equivalent L² value) for different tasks/datasets?

---

> ### Author Response · Authors · 2025-11-22
>
> We sincerely thank you for the insightful feedbacks.
> ## Weakness
> 1. Following your suggestions, we have revised the theorem statements to apply to a broader range of conditions. We have removed the requirements for vanishing initialization and uniform Fisher information.
> 2. We have conducted additional experiments involving larger model scales to address this concern.
> 3. In accordance with your suggestion, we will adjust our wording in the revised manuscript. We aim to align more closely with standard conventions to eliminate any potential ambiguity.
> 4. Our work primarily focuses on investigating the role of noise in NTP. Therefore, we do not address extremely difficult mathematical tasks that strictly necessitate the use of pretrained models. Additionally, since pretrained models are typically trained using NTP, comparing NTP and CTP during subsequent fine-tuning would result in an unfair comparison for both approaches.
> ## Question
> 1. We believe that implicit regularization and explicit regularization can take effect simultaneously. There is already a significant body of literature investigating the impact of weight decay during the NTP pretraining of large models, such as [1,2].
> 2. Yes, we agree with this assessment. This hybrid approach aligns with the standard practices currently used for training modern large language models.
> 3. While the specific definitions of "distributed or compositional representation" can be open to interpretation, we believe that NTP is capable of achieving better performance in real-world tasks. This is largely because real-world corpora are not composed entirely of noise. Our current analysis primarily attributes the robustness to the flatter minima found by NTP.
> 4. Yes, the advantages do persist in larger models. Please refer to our response to Weakness 2 for Reviewer hE2P, where we have provided additional details regarding this matter.
> 5. Quantifying the exact equivalent L2 value for varying tasks and datasets is an interesting but complex challenge. We intend to leave this specific quantitative analysis for future work.
>
> ### Reference
> [1] Hang, Liangkai, et al. "Scalable Complexity Control Facilitates Reasoning Ability of LLMs." arXiv preprint arXiv:2505.23013 (2025).
> [2] Zhang, Zhongwang, et al. "Complexity control facilitates reasoning-based compositional generalization in transformers." arXiv preprint arXiv:2501.08537 (2025).

---

### Official Review · Reviewer_hE2P · 2025-10-28

**Soundness:** 3
**Presentation:** 2
**Contribution:** 2
**Rating:** 2
**Confidence:** 4

**Summary:**

This paper explores the difference between Next Token Prediction (NTP) and Critical Token Prediction (CTP) during pretraining of large language models. The authors argue that NTP’s inclusion of “uncritical” tokens implicitly regularizes the model, similar to L2 weight decay, improving generalization and robustness. They support this claim via synthetic “Anchor Function” experiments, theoretical derivations, and several reasoning benchmarks (e.g., PrOntoQA, CLUTRR, RuleTaker), training models from scratch.

**Strengths:**

- Includes both theoretical reasoning and empirical comparison with CTP.

- Attempts to unify several lines of thought around noise-induced regularization.

**Weaknesses:**

- The theoretical analysis, while intuitively appealing, is not fully rigorous. The core derivation that  $\[L_{\text{NTP}} = \frac{1}{T}L_{\text{CTP}} + \frac{1}{2}\theta^\top I_0 \theta + O(\|\theta\|^3)\] $ relies on overly strong assumptions—such as *uniform Fisher information* and *small-weight initialization*—that are unlikely to hold in deep transformer architectures. The argument only establishes a **local first-order approximation** near initialization, so the claimed equivalence between NTP and L2 regularization does not necessarily persist during full training. Moreover, since $\(I_0\)$ is not isotropic, the resulting penalty is **not true L2 regularization** but a direction-dependent quadratic form. Hence, while the result provides a useful intuition, it should be interpreted as an *analogy* rather than a formal equivalence.


- The paper's main claim—that "NTP is mathematically equivalent to CTP plus weight decay"—is expressed too strongly relative to the presented evidence. The experiments are carefully executed but limited in scale and diversity: all models are small (GPT-2 125M) and trained on synthetic or narrow reasoning datasets. As such, the empirical results do not fully substantiate the universality of the claim. The observed effects may instead be dataset- or scale-specific, and the paper would benefit from more systematic ablations (e.g., across model sizes, initialization scales, or data noise levels) to convincingly support the general conclusion.


- The connection between token-level noise and implicit regularization echoes prior work on SGD noise and dropout; thus, the conceptual novelty may be limited without stronger theoretical or empirical differentiation.

**Questions:**

Refer to Weaknesses

---

> ### Author Response · Authors · 2025-11-22
>
> ## Weakness
> We sincerely thank the reviewer for the thorough review and constructive feedback. We will address Weakness 3 first, followed by the other two points.
> ### Weakness 3
> We would like to clarify the distinction between our work and studies on SGD or label noise. The key difference lies in the specific structure of the noise.
>
> Consider the comparison between NTP and CTP with added label noise. In the latter case, the noise only affects the loss calculation for the critical token. Therefore, the noise impacts only the function $f_S​(\theta,x)$. In contrast, for NTP, the noise acts on every token output by the Transformer model. This means it influences the functions $f_s​(\theta,x), \forall s\in [0,S]$​.
>
> Our work attempts to mathematically formulate the noise loss from NTP as a regularization term for CTP. This derivation is theoretically non-trivial and has not been explored in previous literature. Furthermore, from an empirical perspective, our results regarding the anchor function demonstrate that NTP behaves differently from standard weight decay. We have included a more detailed clarification of this point in the Remark 1 of the revised manuscript.
>
> ### Weakness 1
> The reviewer accurately pointed out the limitations of our theoretical assumptions. Following your suggestions, we have reformulated our theorems under more appropriate assumptions to improve the generalizability of our results. Our analysis shows that in settings without small initialization, the regularization term is controlled by local factors. However, under the specific condition of small initialization, it reduces to simple L2 regularization.
>
> We argue that in the general scenario, while the penalty term is not strictly equivalent to L2 regularization, it still helps in finding solutions that are locally flatter. Such solutions typically correspond to better generalization performance.
> ### Weakness 2
> In response to the reviewer's suggestions and considering the time constraints, we focused our additional verification on the PrOntoQA dataset. This dataset is the primary focus of our discussion. We evaluated performance across different model sizes using standard model architectures. The experimental results align with our expectations. Specifically, NTP consistently outperforms CTP. This trend persists even though we observed that the training speed of CTP accelerates as the model scale increases. We have included these new results in the revised manuscript.

---

> > ### Comment · Reviewer_hE2P · 2025-11-28
> >
> > Thank you for the response. I appreciate the clarifications and the additional experiments. My main concern remains that the models used are still quite small, which limits the strength of the empirical claims. I also understand this is difficult to address within the constraints of the rebuttal. This is why I initially leaned toward a reject.
> >
> > That said, I acknowledge the authors’ efforts and the potential of the direction, so I am raising my score to 4, although I still mind if the paper is accepted (which is why my original score was 2 rather than 4). I encourage the authors to further validate the conclusions at larger scales when resources allow.

---

> > > ### Author Response · Authors · 2025-11-28
> > >
> > > Thank you for your response. We would like to respectfully clarify that we have already addressed the concern regarding model scale in the revised manuscript. As detailed in Section 5.1, **we conducted and reported additional experiments on models ranging from 0.5B to 7B parameters**. These results confirm that our conclusions hold consistent at these larger scales.

---

### Official Review · Reviewer_epoH · 2025-10-30

**Soundness:** 3
**Presentation:** 3
**Contribution:** 3
**Rating:** 6
**Confidence:** 3

**Summary:**

This paper explores why next-token prediction (NTP), which trains on all tokens, outperforms critical-token prediction (CTP), which trains only on the last token. The authors argue that losses on uncritical tokens act as an implicit L2-style regularizer that improves reasoning and generalization. They support this with theory showing NTP equals CTP plus a quadratic regularization term, synthetic experiments where NTP avoids shortcut solutions, and real benchmarks where NTP-trained GPT-2 models show higher reasoning accuracy, better robustness, and flatter minima. Overall, the study concludes that uncritical tokens, though seemingly redundant, play a key role in regularizing models and promoting better reasoning.

**Strengths:**

1.	The paper claims that uncritical tokens act as implicit regularizers. They help models avoid shortcut learning and improve reasoning. This idea is useful and explains why removing tokens during training can sometimes harm generalization.
2.	The paper presents a formal analysis showing that NTP is linked to a quadratic regularizer under certain assumptions. It then tests this idea on several synthetic and real reasoning datasets. This mix of theory and experiments makes the claim stronger and more convincing.
3.	The information-flow visualizations and ablations, such as the NTP shuffle, help show how NTP avoids shortcut learning. They reveal that NTP encourages step-by-step reasoning instead of early merging. These visual results make the findings easier to understand.
4.	The paper shows that NTP-trained models are more robust to embedding noise and achieve flatter minima. This finding connects NTP’s behavior to generalization and SGD theory, supporting the idea that it acts as a regularizer.

**Weaknesses:**

1.	The assumption that the last token is always critical is oversimplified. Datasets like ReCOGS (line 1329-1337) have critical tokens in different position. Using explainability tools (e.g., SHAP or LIME) could better identify important tokens.
2.	The paper mentions token selection methods (RHO-1, Phi-4) but does not evaluate them. It is unclear how partial token dropping affects NTP’s regularization. Experiments across a spectrum—from full NTP to partial selection to pure CTP—would guide practitioners. Intermediate strategies like random token drop or top-k scoring are suggested.
3.	The theory relies on strong assumptions: near-uniform outputs at initialization, uniform Fisher matrix, and quasi-uniform logits at convergence. These may not hold for large-scale pretraining. The authors should discuss this and provide empirical checks (Fisher spectrum, logits uniformity).
4.	All experiments use small GPT-2 models (125M) trained from scratch. This limits realism. The paper misses the opportunity to explore larger and latest models, bigger corpora, or pretrained + fine-tuned setups. Comparing pretrained NTP and CTP on a downstream task would improve practical relevance. Alternatively, test whether the theoretical regularization scales with model or vocabulary size.
5.	The effect of vocabulary size, label frequency, and rare vs. frequent answer tokens on NTP is unexplored. Regularization dynamics are unclear. Tracking parameter norm or Hessian trace across training, with plots for NTP vs CTP, would clarify these effects.

**Questions:**

- Highlight some quantification of the main findings in the abstract and introduction.

- The legend for Figure 6 is confusing and should be clarified.

- Maintain a consistent color scheme for NTP and CTP throughout the paper (check Figures 3, 4, and 6).

- Move a brief explanation of the NTP shuffle experiment (Appendix A.1) into the main text, as it clearly shows structure vs. objective effects.

- Fairness checks showing NTP outperforms CTP under larger token/epoch budgets are only in the appendix. Include key results with actual numbers and a small table for 1×, 2×, and 4× CTP token budgets in the main text.

- Add a short, intuitive sketch in the main text explaining why the second term becomes (1/2) θᵀ I₀θ.

- Appendix D lists compute but omits key hyperparameters: learning rates, schedules, batch sizes, weight decay, and seeds.

---

> ### Author Response · Authors · 2025-11-22
>
> We sincerely thank you for the positive evaluation and constructive feedback. Regarding your weakness and questions:
> ## Weakness
> 1. We agree that ReCOGS does not follow the assumption that the last token is the critical token. After a careful evaluation, we concluded that this dataset is not a reasoning dataset in the strict sense. Instead, it primarily tests the translation from text to logical language. Therefore, we have decided to remove this specific dataset in the revised manuscript.
> 2. We agree that this is a valuable research direction. However, recent studies have already explored balancing prompt loss and response loss (see [1-2]). These works suggest that introducing a certain proportion of prompt loss can improve performance on specific fine-tuning benchmarks. While applying partial NTP during pretraining is an interesting question, it requires extensive resources and time. Consequently, it is difficult to address this fully within the limited timeframe of the rebuttal.
> 3. In the revised manuscript, we have reformulated the theorems to relax the requirement for "small initialization and uniform Fisher information matrix," and rewrite the specific case of small initialization to a separate Proposition. We originally emphasized small initialization condition because, in this scenario, the regularization term could be thought as weight decay, offering intuitive theoretical insights.
> 4. We have extended our tests to larger models. Specifically, we selected the PrOntoQA dataset for detailed analysis and trained models of varying sizes from scratch. The results remained consistent across different model scales (in Section 5.1). Regarding the comparison between pretrained NTP and CTP, we have addressed related work in our response to Weakness 2. Furthermore, since most pretrained models are originally trained using NTP, comparing NTP and CTP directly in downstream fine-tuning tasks could introduce unfairness.
> 5. We acknowledge that token frequency may influence the regularization effect of NTP, as our current analysis relies on the assumption of random noise. In real-world scenarios, token frequencies vary significantly, making prediction difficulty highly variable. We believe this is a complex issue that warrants future exploration. Additionally, it is not immediately clear to us how monitoring the parameter norm or the Hessian trace would directly provide insights into the specific effects of token frequency.
> ## Question
> We sincerely thank the reviewer for the careful review of our paper. We have incorporated and highlighted your suggestions into the revised manuscript. For question 5, we added a brief discussion in Section 7 and for question 6, we have included the intuitive sketch in Remark 2.
> ### Reference
> [1] Huerta-Enochian, Mathew, and Seung Ko. "Instruction Fine-Tuning: Does Prompt Loss Matter?." Proceedings of the 2024 Conference on Empirical Methods in Natural Language Processing. 2024.
>
> [2] Chatterjee, Anwoy, et al. "On the effect of instruction tuning loss on generalization." Transactions of the Association for Computational Linguistics 13 (2025): 1360-1380.

---

> > ### Comment · Reviewer_epoH · 2025-11-24
> >
> > 1, 3, 4) Thank you for addressing these points.
> >
> > 2) It would be helpful if the authors could briefly discuss this in the limitations and discussion section.
> >
> > 3) Thank you for the clarification. I understand that token frequency adds complexity beyond the current theoretical assumptions. My suggestion about tracking parameter norms or Hessian traces was meant as a practical diagnostic tool. These metrics often reveal how implicit regularization evolves during training and could help determine whether token-frequency effects lead to measurable differences in sharpness or parameter growth. Even a small exploratory analysis would make this part of the discussion more concrete. The following works support the idea that monitoring Hessian trace and parameter norms can illuminate implicit regularization:
> >
> >       • Wu et al. show that the trace of the Hessian is tied to linear stability under SGD and relates to parameter norms in certain models [1].
> >
> >       • Gatmiry et al. theoretically and empirically link Hessian-trace minimization to parameter-norm regularization in deep linear models [2].
> >
> > References
> >
> > [1] Wu, Lei, and Weijie J. Su. “The implicit regularization of dynamical stability in stochastic gradient descent.” ICML, 2023.
> >
> > [2] Gatmiry, K., Li, Z., Ma, T., Reddi, S., Jegelka, S., and Chuang, C. Y. “What is the inductive bias of flatness regularization? A study of deep matrix factorization models.” NeurIPS 36, 2023.

---

> > > ### Author Response · Authors · 2025-11-28
> > >
> > > 3) Following your suggestion, we further monitored the evolution of the Hessian and parameter norms using the anchor function (in standard GPT-2 model and revised in **Sec. 6.2**). The results clearly demonstrate that NTP exerts a strong regularization effect, significantly reducing the maximum eigenvalue of the Hessian and driving the model towards flatter regions. In contrast, CTP shows a continuous increase in the maximum eigenvalue, suggesting that the network tends to converge to sharper solutions in the absence of regularization. Consequently, aided by regularization, the NTP model gradually moves away from the initialization to find flatter minima, whereas CTP remains trapped in sharp minima near the initialization point.

---

### Official Review · Reviewer_SXkx · 2025-11-01

**Soundness:** 2
**Presentation:** 3
**Contribution:** 2
**Rating:** 4
**Confidence:** 3

**Summary:**

This paper analyzes the differences between Next Token Prediction (NTP) and Critical Token Prediction (CTP) training approaches. Through theoretical analysis and experimental validation, it proposes that NTP training approximates CTP training with added L2 regularization. Experimental results demonstrate that NTP models exhibit stronger generalization and reasoning capabilities than CTP models, consistently outperforming them across various tasks.

**Strengths:**

The paper conducts thorough theoretical analysis, trains all models from scratch, and compares NTP and CTP model performance alongside loss curve across multiple datasets.

**Weaknesses:**

In practice, NTP serves as the mainstream pretraining approach, while CTP—constrained by challenges like critical token identification—is primarily used for fine-tuning on specific tasks after pretraining. The theoretical approximation proposed in this paper is only applicable to the early stages of training for models trained from scratch. This limitation prevents the paper from adequately addressing the effectiveness of training approaches like CTP following NTP.

**Questions:**

1, It is recommended to include the CTP+WD training combination for comparison in experiments such as Figure 4.
2, Provide performance comparisons between NTP and CTP training approaches using pretrained models on some difficult tasks.

---

> ### Author Response · Authors · 2025-11-22
>
> Thank you for your insightful comments.
> ## Weakness
> We address your concerns regarding the theoretical assumptions and the practical application of CTP (specifically in the fine-tuning stage) in the following three points:
> 1. In the revised manuscript, we have reformulated the theorems to relax the requirement for "small initialization and uniform Fisher information matrix," and rewrite the specific case of small initialization to a separate Proposition. We originally emphasized small initialization condition because, in this scenario, the regularization term could be thought as weight decay, offering intuitive theoretical insights.
>
> 2. We agree that the "CTP following NTP" paradigm (fine-tuning) is a valuable direction for research; however, it falls outside the scope of the current study. Our primary objective in this paper is to investigate whether noise within the corpus impairs model reasoning abilities reasoning abilities during the training from scratch phase.
>
> 3. Although our focus is on pretraining, we acknowledge the importance of the fine-tuning scenario. Existing observations confirm that inappropriate noise introduced by NTP can hinder the effectiveness of fine-tuning on pretrained models. Our supplementary tests demonstrate that CTP significantly outperforms NTP in this context.
>
>     - Experiment: We fine-tuned an OPT-350M model on the MetaMathQA dataset and evaluated it via autoregressive generation on GSM8K.
>
>     - Results: We observed that CTP exhibits faster convergence in testing accuracy compared to NTP. Ultimately, CTP achieved an accuracy of 0.25, whereas NTP only reached 0.22.
>
>     - Recent work indicates that an appropriate "prompt loss" can aid generalization (see [1,2]), which adds nuance to the discussion of objective functions in fine-tuning.
>
> ## Question
> To address your question, we have added the accuracy results for both GPT-2 and Qwen3-8B models on the PrOntoQA 2-hop task (Figure 4). Our experiments indicate that CTP combined with Weight Decay (CTP+WD) yields significant improvements in larger models. However, for smaller models, there is no substantial difference compared to standard CTP.
>
> Regarding more complex tasks, these cannot be learned effectively by training from scratch. Relying on pretrained models for these tasks would introduce unfairness into the comparison between NTP and CTP. We have already discussed this issue in our response to Weakness point 3, so we will not repeat those details here.
>
> ### Reference
> [1] Huerta-Enochian, Mathew, and Seung Ko. "Instruction Fine-Tuning: Does Prompt Loss Matter?." Proceedings of the 2024 Conference on Empirical Methods in Natural Language Processing. 2024.
>
> [2] Chatterjee, Anwoy, et al. "On the effect of instruction tuning loss on generalization." Transactions of the Association for Computational Linguistics 13 (2025): 1360-1380.

---

### Author Response · Authors · 2025-12-02
**Global Response: Summary of Revisions/Response to Reviewer Feedback**

We sincerely thank all the reviewers for their time and constructive feedback. We would like to reiterate our central takeaway: **the noise present in pretraining corpora benefits a model's acquisition of reasoning capabilities by acting as an implicit regularizer.** We selected various corpora and model sizes to verify the universality of this conclusion, and also established theoretical analysis for simple models.

Below we detail the major revisions/responses following the rebuttal phase：

| Purpose | Reviewer(s) Addressed | Summary |
| --- | --- | --- |
| Theoretical assumptions are only applicable to the early stages of training | **SXkx, epoH, hE2P, aZhb** | We reformulated the theorems to relax the assumptions. We rewrite the specific case of small initialization to a separate Proposition |
| Address the effectiveness of training approaches like CTP following NTP | **SXkx, epoH, aZhb** | The question is out of scope of our topic and we provide Recent work 1,2] indicates that prompt loss can aid generalization in fine-tuning. |
| All experiments use small GPT-2 models (125M) trained from scratch which limits realism. | **epoH, hE2P, aZhb** | we selected the PrOntoQA dataset for detailed analysis and trained models of varying sizes from scratch. The results are updated in revised manuscipt. |
| Strength analysis by monitoring parameter norm or the Hessian trace | **epoH** | We add relevant analysis in **Sec. 6.2** and demonstrate that NTP exerts a strong regularization effect. |
| The noise regularization is similar to SGD or dropout, where novelty is limited. | **hE2P** | We clarify the difference between NTP and “SGD or label noise”: Former affacts $f_S(\theta,x)$ but latter affacts $f_s(\theta,x), \forall s\in [0,S]$. This derivation is theoretically non-trivial and has not been explored in previous literature. |
| The research on more difficult datasets | **aZhb** | We clarify that pretrained models are typically trained using NTP, comparing NTP and CTP in diffult math problems would result in an unfair comparison. |

During the review process, the most active reviewer, epoH, took a positive view of the paper, affirming its topic and its meticulous and persuasive experimental design, while also offering a series of constructive comments. Unfortunately, there was no further discussion from other reviewers before November 27.

We thank everyone for their help in shaping the paper to be in better form. The feedback has genuinely strengthened our work and helped us present our contributions with greater clarity and rigor. We remain fully available for any further questions and we hope the revised manuscipt have adequately addressed all concerns raised.

---

### Meta-Review · Area_Chair_UhiN · 2026-01-05

**Summary:**

The concerns raised by the reviewers are primarily as follows:

1. The paper studies next token prediction (NTP) vs. critical token prediction (CTP) for **pretraining**, and reveals that the former is important in offering regularization benefits.
However, as raised by Reviewers SXkx and aZhb, the former has been the dominant pretraining objective for LLMs---to be fair, I'm not aware of any approach that uses CTP for pretraining.

2. There are a few more concerns on the clarity of the presentation (aZhb) and the oversimplified assumptions in the CTP settings (epoH, hE2P, aZhb).

**Reviewer Concerns:**

I think Concern 2 is partially addressed in the rebuttal and revisions; however, Concern 1 remains unaddressed. I don't think this paper adds valuable insights to the community of pretraining.

**Reviewer Scores:**

I don't think Reviewers SXkx, aZhb, and epoH will change their ratings. The concerns by the first two reviewers are not adequately addressed, and Reviewer epoH has already been positive. There is a considerable chance for Reviewer hE2P to raise their rating to 4, but this does not cross the acceptance threshold.

---

### Decision · Program_Chairs · 2026-01-26

Reject